# Late Acheulian Jaljulia – Early human occupations in the paleo-landscape of the central coastal plain of Israel

Maayan Shemer[1,2]*, Noam Greenbaum[3], Nimer Taha[4], Lena Brailovsky-Rokser[5], Yael Ebert[6], Ron Shaar[6], Christophe Falgueres[7], Pierre Voinchet[7], Naomi Porat[8], Galina Faershtein[8,9], Liora Kolska Horwitz[10], Tamar Rosenberg-Yefet[11], Ran Barkai[11]

1 Archaeological Research Department, Israel Antiquities Authority, Jerusalem, Israel, 2 Ben Gurion University of the Negev, Beer Sheva, Israel, 3 Department of Geography and Environmental Studies, University of Haifa, Haifa, Israel, 4 Dr. Moses Strauss Department of Marine Geosciences, L.H. Charney School of Marine Sciences, University of Haifa, Haifa, Israel, 5 Israel Antiquities Authority, Rockefeller Archeological Museum, Jerusalem, Israel, 6 The Fredy & Nadine Herrmann Institute of Earth Sciences, The Edmond J. Safra Campus, The Hebrew University of Jerusalem, Givat Ram, Jerusalem, Israel, 7 UMR 7194, CNRS, MNHN, UPVD, Sorbonne Universités, Institut de Paléontologie Humaine, Paris, France, 8 Geological Survey of Israel, Jerusalem, Israel, 9 Department of Earth and Planetary Sciences, Weizmann Institute of Science, Rehovot, Israel, 10 National Natural History Collections, Hebrew University, The Edmond J. Safra Campus, Givat Ram, Jerusalem, Israel, 11 Department of Archaeology and Near Eastern Cultures, Tel Aviv University, Tel Aviv, Israel

* Shemerma@hotmail.com

**Data Availability Statement:** All relevant data are within the manuscript and its Supporting Information files.

## Abstract

The Lower Paleolithic Late Acheulian in the Levant marks a fascinating chapter in human cultural and biological evolution. Nevertheless, many aspects of the Late Acheulian are still undeciphered, hindered by the complex nature of each site on the one hand, a scarcity of wide, multidisciplinary studies on the other, and by difficulties in obtaining absolute chronology for this timeframe. Therefore, subjects such as human subsistence strategies and modes of adaptation, regional diversity, and the possible existence and nature of interactions between hominin groups are largely understudied. The discovery and study of Jaljulia, a large-scale Late Acheulian site at the central Coastal Plain, Israel, add valuable insights to the research of this chapter in human history. Considered to represent recurrent occupations at a favored, water and flint-rich setting, the site has provided extensive lithic assemblages obtained from several localities. Absolute chronology places the human activity on-site at roughly 500–300 ky (and possibly even later), which is suggested to be divided into several main occupation phases. Geomorphological and sedimentological analyses show a change in environmental conditions, from aeolian sand deposition and overlying Hamra soil during the Middle Pleistocene to high energy fluvial regime which transported large gravels in a north-south paleo-channel. Wetland environments, correlating to the human activity on site, developed later due to higher sea levels and a coastline shifts to the eastward, which caused a blockage of the Yarkon stream corridor to the sea by marine sand. In this paper we present results of the study of the site, including geomorphological formation and post-depositional processes, absolute chronology, lithic and faunal analyses. The site's extensive lithic assemblages are currently under study and future investigations are expected to shed more light on the technological nature of Late Acheulian Jaljulia.

**Funding:** The excavations at Jaljulia were founded by the Israel Land Authority as a part of pre-land development procedures. Post-excavation research is funded by the Israel Science Foundation (grant no. 321/19; RB). https://www.isf.org.il/#/ and by the Israel Antiquities Authority (project no. 108149; MS). http://www.antiquities.org.il/default_en.aspx The funders had no role in study design, data collection and analysis, decision to publish, or preparation of the manuscript.

**Competing interests:** The authors have declared that no competing interests exist.

## Introduction

The term "Late Acheulian" is often used to reflect chronological and behavioral aspects associated with advanced stages of the long and persistent Lower Paleolithic Acheulian cultural complex in the Levant (e.g., [1–3]). To date, dozens of Late Acheulian sites and find spots are reported in the Southern Levant. Their wide geographic spread indicates human activity in diverse environments, including the Levantine coastal plain, the Mediterranean inland, the Galilee, the Golan Heights and the arid regions of the Negev Desert, Southern Jordan and the Arabian Desert [4–21]. Nevertheless, our understanding of the human mode of adaptation, cultural and biological transformations and technological developments practiced in the course of this phase of human history is not detailed enough. Only a handful of Late Acheulian sites have been excavated in the past decades, using advanced excavation and documentation techniques, that include a multidisciplinary research program (e.g. [17, 19, 22–24]). Therefore, some aspects, such as absolute chronology of the Late Acheulian as well as human subsistence strategies, adaptation patterns and technological diachronic changes remain largely understudied.

Late Acheulian sites are often associated with the nearby presence of an ancient water body, characterized in many cases by recurrent visits to a favorable locality indicated by sequences of overlying occupation horizons or by the presence of multiple, adjacent find spots (e.g., [8, 10, 19, 23–26]). In several of the excavated sites, rich archaeological layers were exposed that were suggested to represent a palimpsest of several occupation events [6, 8, 17, 27, 28].

In terms of Late Acheulian material culture, the use of flint as a preferred stone type for tool production is demonstrated, in contrast to the more frequent use of basalt and limestone for specific tool categories in older Acheulian industries (e.g., [3, 9, 29], and see [30] for insights about the infrequent use of basalt in the Late Acheulian site of Ma'ayan Baruch). Late Acheulian lithic assemblages are characterized by large-scale flake production accompanied by shaped flakes and typical 'core' implements such as bifaces and chopping tools. Core technology share attributes with earlier Acheulian industries, demonstrating flake production from globular cores and cores-on-flakes while using one, two or more striking platforms (e.g., [6, 8–10, 25, 31–35]). Nevertheless, lithic studies have suggested techno-typological inter-assemblage variability, implying the probable effect of chronology, geography and the nature of the activities performed, on the lithic diversity within Late Acheulian sites (e.g., [1, 3, 24, 36–41]).

One of the noteworthy characteristics of Late Acheulian lithic industries in the Levant is the production of pre-determined target flakes from prepared cores (also termed Hierarchical cores or proto-Levallois cores), and/or from discarded, recycled handaxes [42]. The production of pre-determined blanks from prepared cores was identified in earlier Acheulian industries, for example at the site of Gesher Benot Ya'aqov [34]. However, it seems to be more common in the Late Acheulian as it appears in numerous associated Levantine assemblages. This production technique is perceived by many as the establishment of volumetric knapping approaches during Late Acheulian [6, 9, 10, 25, 31, 32, 43]. The adoption of volumetric conceptions in flintknapping and the production of predetermined blanks from prepared cores have become the subject of much attention, as its possible role in the development of the later, Middle Paleolithic Levallois method has fundamental implications for our understanding of ancient hominin behavior, learning capabilities and adaptability (e.g., [44–47]).

One of the main issues in the study of the Late Acheulian relates to its position at the end of a long, technologically persistent cultural complex, which lasted in the Levant over one million years. The disappearance of traditional Acheulian lithic industries from the Levantine record at the end of the Lower Paleolithic period marked the onset of new cultural and technological traditions e.g., the Acheulo-Yabrudian, the Mousterian). However, the nature and chronology

of such transitions are still largely undeciphered, as well as the degree of interaction and possible cultural influences between diachronic and synchronic human groups.

Difficulties in acquiring absolute ages further add to the ambiguity. To date, a wide chronological framework is assumed for the Late Acheulian, based on radiometric dating of volcanic strata from the Golan heights [9], a series of optically stimulated luminescence (OSL) and electron spin resonance (ESR) ages (e.g., [19, 20, 23, 40, 48]), thermoluminescence (TL) ages [49] and techno-typological aspects of the lithic assemblages (e.g., [1, 3]). Late Acheulian occurrences are considered younger than ca. 800 ky. The Acheulo-Yabrudian Cultural Complex (henceforth AYCC), marking the end of the Lower Paleolithic period in the Levant, was mostly dated between 420/400 and 250/200 ky (e.g., [50–54]). Thus, the Late Acheulian in the Levant is mostly considered to be bracketed between 800 to 400 ky. However, this chronological framework has been somewhat challenged in recent years, as some Late Acheulian sites yielded radiometric ages younger than 400 ky. For example, while the Late Acheulian sites of Kefar Menahem and Nahal Hesi provided ages older than 400 ky [40, 55], the site of Revadim provided preliminary indications of an age older than 500–300 ky [19] and the sites of Holon (Israel), Shishan Marsh 1 (Jordan) and Oumm Qatafa (D2) yielded much later chronologies, ranging between ~300 and ~200 ky [20, 23, 48]. These younger ages were used to suggest a Late Acheulian presence in the Levant up until the establishment of Middle Paleolithic industries in the region (e.g., [23, 48]). However, this approach is viewed with caution, as most scholars view an autochthonic cultural continuum between the Late Acheulian and the AYCC. Based on the plethora of evidence available, the AYCC was suggested to have been developed from Late Acheulian industries, while embracing some of its technological hallmarks (e.g., bifaces, lithic recycling) and introducing a set of new innovative technologies and behaviors (e.g., systematic blade production, Quina scraper production, habitual use of fire, cooking etc.; e.g., [56, 57]).

Thus, a more comprehensive understanding of the chronology of the Late Acheulian is still to be desired, and the nature of technological and behavioral transformations that took place during the terminal Lower Paleolithic in the Levant deserves special attention.

In this paper we present the discovery of the large-scale Late Acheulian site of Jaljulia, located in the central coastal plain of Israel. The site was discovered during routine inspections of the Israel Antiquities Authority prior to a major construction operation. Two seasons of salvage excavation followed, resulting in six months of field work and data collection oriented towards a multidisciplinary study of the site and of early hominin behavior in the paleo-landscape. Preliminary chronological estimations led to the association of the site with the Late Acheulian. Here we provide detailed information about the stratigraphy, sedimentology, and absolute chronology of the site, alongside a suggested reconstruction of the paleoenvironment, formation and post-depositional processes observed on site. In addition, preliminary analyses of the flint industry are presented, as well as a study of the faunal assemblage.

## Materials and methods

### The site and excavation

The site of Jaljulia is located in the central coastal plain of Israel, ca. 9 km from the current Mediterranean shoreline (Fig 1). The site was discovered in the course of test trenches inspection conducted in the fall of 2016, prior to a large-scale construction project which was intended to include massive earthworks. Upon reaching a depth of 1.5–5.5 m, numerous flint artifacts appeared, indicating the presence of rich archaeological deposits, found in an area of ca. 10 dunam (1 hectare; Fig 1). The discovery led to the initiation of a large-scale salvage project (conducted between June and December 2017; license no. A-8000/2017), during which six

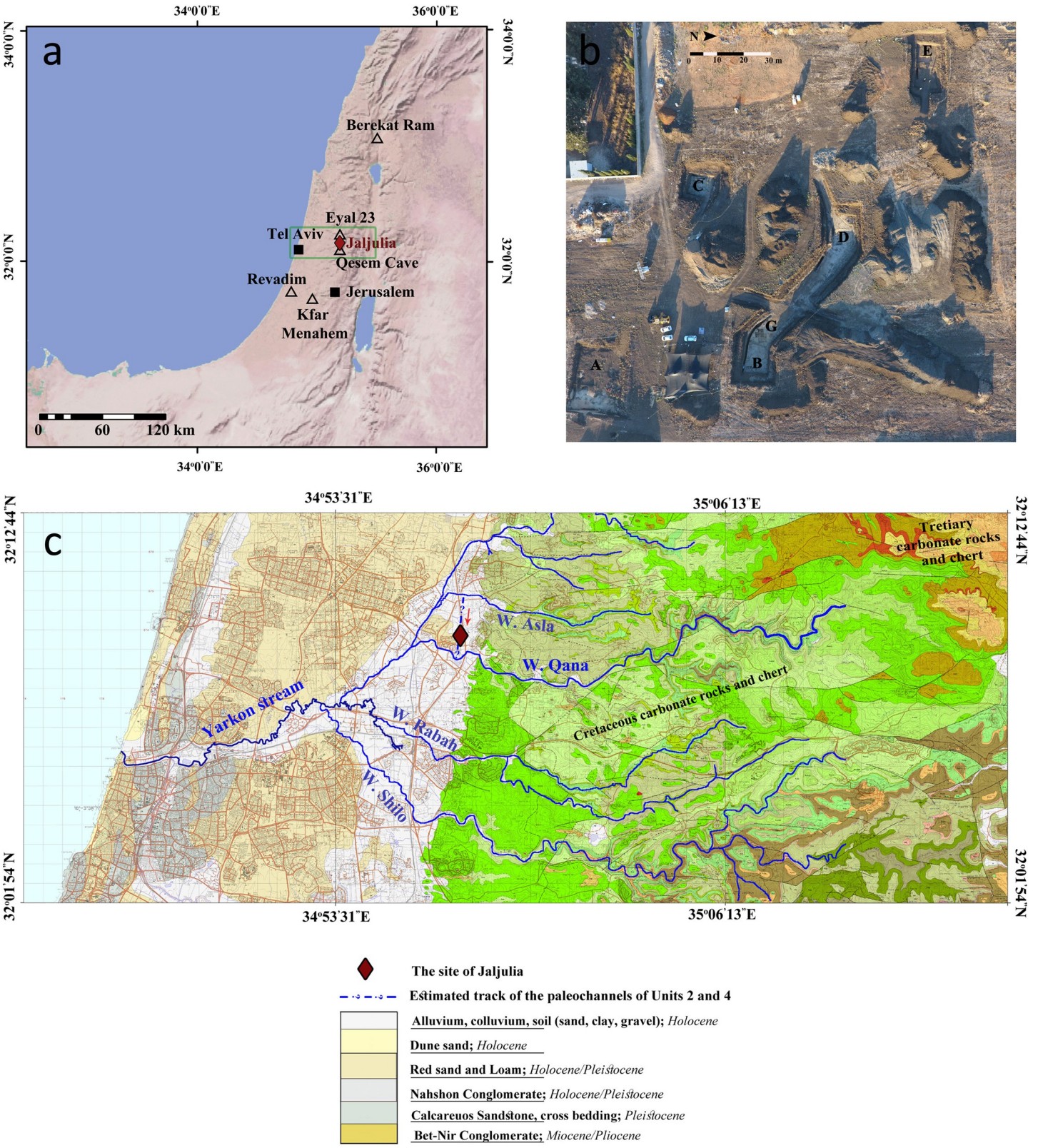

**Fig 1. Location and geological setting of the site of Jaljulia.** a) Location of the site of Jaljulia and main sites mentioned in this paper (base-map source: US National Park Service). Green frame marks the area presented in Fig 1c; b) Geographic distribution of the excavated localities in Jaljulia; c) Geological setting of the site of Jaljulia, with marks of the main streams associated with the formation processes of the site (base-maps provided by the Geological Survey of Israel, created by [58–60]).

localities were excavated, covering a total area of approximately 80 m$^2$ (Areas A–E, G; Figs 1B and S1–S6). The depth of the archaeological deposits, covered by more than three meters of sediments in most of the excavated areas, required heavy machinery to remove the overlying sediment as well as to create stepped walls where manual excavation was to be conducted, in order to prevent collapses and in accordance with safety considerations. In addition, a geological trench was dug in each area using a JCB backhoe, in a twofold effort: 1) to provide as much data as possible about the geomorphology of the site, including the collection of sediment samples for absolute dating, and 2) to correlate between the different excavated localities. For the most part, these trenches were located at the southern edge of each area, adjacent to the stepped wall. In Area A, located at the southeastern part of the site (Fig 1B) a second trench was dug along the eastern boundary of the excavated area in order to better understand the local stratigraphy.

In each area, 6–25 m$^2$ were excavated, applying a general grid of 1 m$^2$ squares, following a north-south orientation. Upon reaching the archaeological layer each square was further divided into four sub-units and excavated in 5–10 cm spits. Specific items, such as bifacial tools and cores were photographed before removal, and their location was recorded in three coordinates. All excavated sediments were dry sieved using a five mm mesh. Sediments from Area G were further subjected to wet sieving using a two mm mesh, due to high density of small finds (<2 cm). As the archaeological finds were, in most cases, deposited on top of concentrations of flint nodules and cobbles, preliminary sorting of the flint items collected during the excavation was conducted on site, separating artifacts from unmodified nodules and natural debris. Unmodified flint was kept only when it originated from a designated "control square" in each area. Documentation was conducted using the Digital Archaeology and National Archive (DANA) databasing program. Drone technology was used for photography and photogrammetric recording.

## Stratigraphy sedimentary units and soils

The definition of sedimentary units and soil stratigraphy was determined based on a detailed study of seven geological sequences. In each sequence, all the observed sedimentological units were sampled. Sedimentary documentation was conducted following Gardiner and Dackombe [61], color attribution determined using Munsell soil color chart. Soil description is in accordance with Birkland et al. [62] and soil definition is based on Dan et al. [63]. These were correlated to the map of soils published by Dan and Raz [64]. Chemical and mineralogical analyses of the sediments were performed at the laboratories of the School for Marine Sciences, University of Haifa, headed by Dr. N. Waldman. The analyses were performed on the fine fraction <2 mm and included: (1) Major elements using XRF; (2) Inorganic Carbon; (3) Grain-size distribution; and (4) Qualitative mineralogy using XRD. The sedimentary units in the seven sequences were compared, correlated, and integrated into a generalized stratigraphic sequence for the entire site (Fig 2).

## Paleomagnetic stratigraphy

Within the framework of the study of magneto-stratigraphy, 20 oriented sediment samples were collected from five horizons in the geological section of Area B. The sampled horizons were labeled GAL1–5, where GAL1 represented the deepest sampled horizon and GAL5 the highest. GAL1–3 underly the archaeological layer and GAL4–5 overly it (Fig 2). From each horizon three oriented samples were collected for Alternating Field (AF) demagnetization experiment and an additional sample was collected for thermal demagnetization experiments.

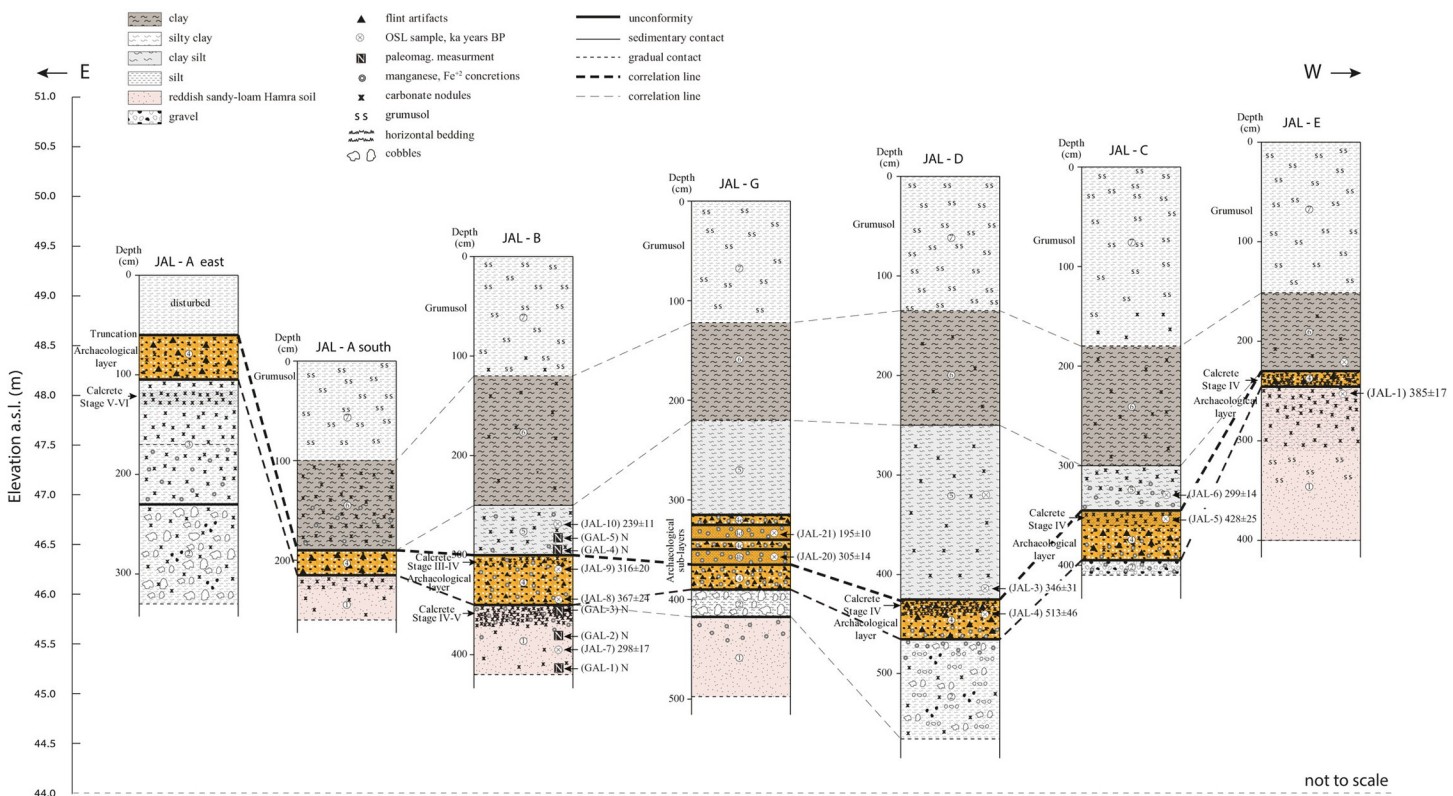

**Fig 2. The stratigraphy of the site of Jaljulia as displayed in all sampled sections, as well as the sampling locations for absolute dating and for paleomagnetic stratigraphy.**

### Field sampling for absolute dating

Sediment samples for absolute dating were collected from the geological sections of Areas B, C, D, E and G. No dating samples were collected from Area A, due to high cementation of the archaeological horizon and an observed truncation of the ancient stratigraphy by younger sediments (see below) in that locality. A total of 11 samples were collected from Units 1, 4 and 5 (JAL 1, 3–10, 20–21; Fig 2; see detailed description of these units below) by drilling horizontally into the exposed sections, after cleaning the surface from material that has been exposed to sunlight. For the sampling of Unit 1, vertical drilling was conducted into the exposed top of the unit in Area E. An opaque cover was used to prevent sample exposure to sunlight. Where drilling was not possible, either because the sediment was too indurated or rich in coarse gravel, samples were collected by scraping sediment into light-tight bags under the opaque cover. An additional large, representative sample (~1 kg) from each location was collected for dose rate determinations. Upon reaching the lab, five samples (JAL 3, 4, 5, 6, 9) were divided into two parts, one to be used for luminescence dating of quartz and potassium (K) feldspars grains, and the other for age estimation of quartz grains by electron spin resonance (ESR). Burnt flints were identified by the naked eye in most excavation localities, and samples of burnt flints will be submitted for Thermoluminescence (TL) dating in the near future.

### Luminescence dating

**Mineral separation.**   Quartz and feldspar in the size range of 90–125 μm were extracted from the sediment as follows: after wet sieving to the desired grain size, carbonates were

dissolved by soaking overnight in 8% HCl, followed by rinsing and drying. The sample was then passed through a Frantz magnetic separator with a current of 1.4 A on the magnet [65]. About 3 gr from the non-magnetic fraction was further etched with 40% HF for 40 min followed by an overnight soak in 16% HCl to remove any fluorides which may have precipitated. For seven samples, K-feldspars (KF) were extracted from ~5 gr of the un-etched non-magnetic fraction using a single-step density separation at a 2.58 gr/cm$^3$, followed by etching of the lighter, KF-bearing fraction with 10% HF for 10 min [66]. This fraction had proved to be highly enriched in KF [66]. After thorough rinsing and drying the quartz and KF were measured.

**Luminescence measurements.** Preliminary measurement of the equivalent dose (De) of quartz using the optically (blue) stimulated luminescence (OSL) signal and the single aliquot regenerative dose (SAR) protocol showed that the OSL signal is saturated, and any age obtained from that signal would be underestimated. Further De measurements used the thermally transferred (TT) OSL signal on quartz and the post infrared-infrared protocol at 250°C and 290°C (pIR-IR$_{250}$ and pIR-IR$_{290}$) signals on KF, two signals capable of extending the range of luminescence dating [67]. Table 1 gives the measurement protocols and parameters used for data analyses. The TT-OSL signal is measured after depleting the main OSL and applying a second preheat treatment to induce a transfer of charge into the main quartz OSL trap [68] (Table 1). This signal saturates at much higher doses and is suitable for dating Middle Pleistocene sediments [69]. The SAR protocol of Porat et al. [70] was used, modified so that the cleaning step at the end of each measurement cycle was at 350°C instead of 310°C [71].

The De for the KF samples was measured using the pIR-IR$_{250}$ and pIR-IR$_{290}$ SAR protocols as in Buylaert et al. [72]. This protocol first depletes the IR signal at 50°C, a signal which is known to fade and provides underestimated ages [73], followed by a second measurement of the IR signal at elevated temperatures [74] (Table 1). These high temperature signals saturate at high doses and are very promising for dating Middle Pleistocene sediments [67]. Both temperatures were used as there is evidence that while the pIR-IR$_{290}$ fades less than the pIR-IR$_{250}$ signal

**Table 1. Measurement protocols and details of quartz TT-OSL and feldspar pIRIR290 signals.**

|  | TT-OSL | pIR-IR$_{290}$ |
|---|---|---|
| **Discs/cups** | Aluminum discs | Stainless steel cups |
| **Aliquot size** | 5–8 mm | 1–2 mm |
| **Stimulation source** | Blue LED's (470 nm) | Infrared diodes (870 nm) |
| **Detection filters** | 7.5 mm U-340 | Schott BG-39 & Corning 7–59 |
| **Signal & background** | First 1 s & last 5 s | First 1 s & last 10 s |
| **Fit** | Exponential or quadratic | Exponential + Linear |
| **Step**[*] |  |  |
| **1** | Give β dose | Give β dose |
| **2** | TL at 260°C for 10 s | Preheat at 320°C for 60 s |
| **3** | Blue stimulation at 125°C for 300 s | IR stimulation at 50°C for 200 s |
| **4** | TL at 260°C for 10 s |  |
| **5** | Blue stimulation at 125°C for 100 s (*Lx*) | IR stimulation at 290°C for 200 s (*Lx*) |
| **6** | Test dose (2.2 Gy) | Test dose (30 Gy) |
| **7** | TL at 220°C for 10 s | Preheat at 320°C for 60 s |
| **8** |  | IR stimulation at 50°C for 200 s |
| **9** | Blue stimulation at 125°C for 100 s (*Tx*) | IR stimulation at 290°C for 200 s (*Tx*) |
| **10** | Heat at 350°C for 100 s | IR stimulation at 350°C for 300 s |

[*]In the first measurement of the natural signal, the given β dose in Step 1 is 0. The highlighted steps were used for constructing the Dose Response Curve (DRC). Modified from Faershtein et al. [82].

[74], it also bleaches more slowly than the pIR-IR$_{250}$ signal, and ages calculated from it might be over-estimated due to remaining residual signal [75]. Nonetheless, since the sedimentary environment is that of a large riverbanks and sands, it could be expected that the pIR-IR$_{290}$ would be well bleached. Rates of fading were measured for the pIR-IR$_{250}$ signal using the approach of Auclair et al. [76] and the measurement protocol of Faershtein et al. [75].

**Dose rates.** The sediment samples collected from the same sampling locations were dried, crushed, homogenized and powdered, and the content of the radioactive elements was measured using inductively-coupled plasma (ICP) mass spectrometry (MS) for U and Th, and ICP optical emission spectroscopy (OES) for K. Internal K-contents within the KF were estimated at 12±0.5%. Lifetime water contents were evaluated at 10±3%, based on the sandy and consolidated nature of the sediment, and variability across seasons and the geological timescale. Cosmic dose rate was calculated from current burial depth, since initial age calculations showed that the overlying Unit 5 is not much younger than the archaeological Unit 4.

## ESR dating

Analyses were conducted on 100–200 μm quartz grain, using the aluminium and titan centers (Al-center, Ti-Li center). Quartz was prepared and separated by N. Porat according the same procedure as used for luminescence analyses (see above).

Irradiation was performed using a $^{137}$Cs gamma source (Gammacell) at CENIEH, Burgos, Spain, with a dose rate of 380 Gy/h. The samples were separated into 9 aliquots and given doses ranging between 150 and 12,500 Gy. For each sample, one aliquot was bleached by exposure to light from a solar simulator (Dr Honhle SOL2). The light intensity received by each aliquot range between 3.2 and 3.4×10$^5$ Lux and samples were illuminated for 1600 h. One other aliquot was kept as naturally irradiated sample. D$_e$ was determined by the multiple aliquot additive (MAA) dose method.

ESR measurements were performed close to the liquid nitrogen temperature (ca. 107 K) with a Bruker EMX spectrometer using the experimental conditions proposed by Voinchet et al. [77]. The Al signal intensity was measured between the top of the first peak g = 2018 and the bottom of the 16th peak g = 2002 of the part of the main hyperfine structure [78]. The Ti-Li center was measured between the base of peak at g = 1.913 and the baseline [79].

Each aliquote was measured three times after a ca. 120˚ rotation in the ESR cavity, in order to evaluate the angular dependence of the signal due to sample heterogeneity. This procedure was repeated at least three times over different days to check measurement and D$_e$ variability. The bleaching rate δbl (%) was determined for each sample by comparison of the ESR intensities of the natural (Inat) and bleached (Ibl) aliquotes.

$$\delta bl = ((Inat - Ibl)/Inat) \times 100)$$

Equivalent doses were then determined by fitting an exponential + linear function through the mean ESR intensities [80, 81] with Microcal OriginPro 8 software and 1/I$^2$ weighting.

I is the intensity of the sample for an irradiation dose D, Isat is the saturation intensity, μ is the radiation sensitivity coefficient of the sample and D$_e$ is the equivalent dose.

The Ti-Li growth curve was calculated from the seven first irradiation doses, the two highest (8 000 and 12 500 Gy) showing saturation and a decrease of signal intensity.

**Dose rates.** The dose rates were calculated from the radionuclide activities of the raw sediment obtained by gamma-ray spectrometry (Natural History National Museum Laboratory, Paris–MNHN) with an Ortec high purity low background Germanium (HPGe) detector (S1 Table). Age calculations were performed using the dose-rate conversions factors from Guérin et al. [82], a k-value of 0.15 ± 0.1 [83], alpha and beta attenuations were estimated for the

selected grain sizes from the tables of Brennan [84, 85], water attenuation formulae from Grün [86] and a cosmic dose rate calculated from the equations of Prescott and Hutton [87] including latitude and altitude corrections. Internal dose rate was considered as negligible due to the low radionuclides contents usually measured in quartz grains [88, 89]. Analytical uncertainties and weighted ages calculated using Isoplot 3.0 software [90].

# Results

## Stratigraphy, sedimentary units, and soils

The geological sections of Jaljulia represent ~5 m of sedimentary sequences, encompassing a wide range of deposits such as aeolian sand, coarse and fine fluvial deposits, inland wetland environments and soils. The sequence was divided into seven units, presented here from the oldest (Unit 1) to the youngest (Unit 7). In Area A, the upper part of the sequence that was exposed in the eastern section (above Unit 4; Fig 2) was truncated, probably due to recent agricultural activity.

The results of the chemical analyses are presented in Table 2 and Fig 3. (For details see S2 and S3 Tables). In general, the main components in most sedimentary units are quartz (expressed as $SiO_2$) assumed to be aeolian in origin and includes sand and dust (58–77%), and Ca-carbonate mineral (calcite, expressed as $CaCO_3$) that is either related to dust or from local sources. Units, 1, 3 and 4 that are overtopped by calcic soils are extremely rich in calcite (63–78%). The aluminium oxide concentrations, 11.6–18%, indicate the presence of considerable amounts of clay minerals. With iron oxides contents of 2.1–7%, they suggest chemical weathering and pedogenic processes in humid/wet environment. Low aluminium oxide concentrations characterize the calcic soils, indicating drier conditions. A similar trend was found for iron oxides, though the values are lower.

The texture of all sedimentary units range between sandy-loam to clay. Clay contents are up to 75%. The common clay minerals for all units are mainly smectite (montmorillonite), kaolinite, and some chlorite.

**Unit 1** is a very hard and compacted, reddish-brown, silty-clay Hamra soil overlying yellowish sand. Its upper part is fine, rich in clay, and can be prismatic in structure. Its lower part is composed of yellowish, massive, compacted sand, which is the parent material for the Hamra soil. The contact between the upper and lower parts is gradual and probably represents the boundary between the sand and the overlying Hamra soil. At the northern part of the site (Areas G, D) Unit 1 is deep, particularly in comparison to its elevation at the western part (Area E) (Fig 2). In Area G the sediments contained gley mottling as well as high concentrations of iron and manganese oxides in the form of concretions, indicating hydromorphic conditions. Large carbonate nodules (ca. 5 cm) and dense carbonate mottling are frequent within the upper part of the unit, forming in places a stage III+–IV- calcrete.

**Table 2. Summary of chemical analyses results shown as oxides or carbonates (for Ca), arranged by unit.** For full data see S2 and S3 Tables.

| Sample # | Unit | SiO₂ | CaCO₃ | Al₂O₃ | Fe₂O₃ | MnO | *CaCo3 (%) | Ca / Si | Si /Al |
|---|---|---|---|---|---|---|---|---|---|
| 13 | 1 | 58.22 | 8.6 | 17.94 | 4.42 | 0.30 | 0.5 | 1.26 | **2.87** |
| 4 | 2 | 57.84 | 9.3 | 19.13 | 4.55 | 0.16 | 4.75 | 0.14 | **2.67** |
| 9,12 | 3 | 13.0–18.0 | 70.7–78.0 | 6.5–8.2 | 0.5–0.8 | 0.04–0.05 | 73.0–85.7 | 3.37–5.14 | **1.76–1.93** |
| 3,8,10 | 4 | 35.2–71.4 | 3.0–46.2 | 11.6–14.9 | 2.1–5.0 | 0.1–1.73 | 1.75–41.2 | 0.05–1.12 | **2.68–4.46** |
| 1, 11 | 4b, 4d | 60.0–62.9 | 1.9–3.5 | 14.6–17.5 | 6.5–7.0 | 1.33–1.85 | 0.08–1.5 | 0.03–0.05 | **3.04–3.81** |
| 2,7,15 | 5 | 69.4–76.7 | 3.0–8.6 | 12.6–14.9 | 2.9–4.1 | 0.08–0.26 | 0.42–7.67 | 0.03–0.1 | **4.12–5.39** |
| 5,6,9,12,14 | calcretes | 13.0–25.4 | 62.7–78.0 | 6.5–8.9 | 0.53–1.6 | 0.04–0.32 | 64.5–85.7 | 2.12–5.14 | **1.76–2.96** |

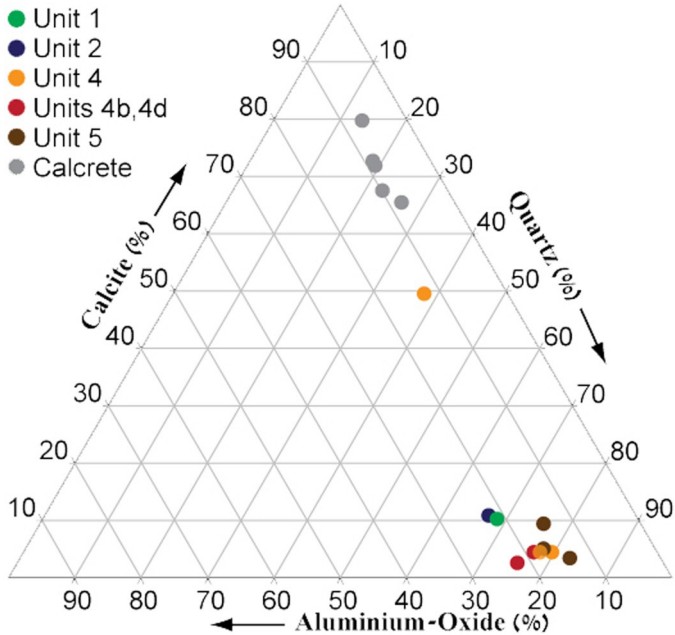

**Fig 3. Ternary diagram showing the proportions of the major components in all sample.** For full analyses see S2 and S3 Tables.

Unit 1 is quartz-rich (58%), with considerable amounts of aluminium and iron oxides (18% and 4%, respectively, Figs 2 and 3 and Table 2), and poor in calcite, suggesting a well-drained soil. The moderate Si/Al ratio (2.87) and the significant amounts of aluminium oxides indicate progressive chemical weathering and presence of significant amounts of clay minerals. The relatively high iron oxides and the soil reddish-brown color suggest aerobic conditions. The top of the unit is covered by stage III-IV calcic soil as indicated by the accumulation of calcite, implying limited rainfall amounts.

**Unit 2** is a fluvial unit, clast or matrix supported coarse gravel unit (50–60% matrix), 0.8–1.0 m thick. Gravels are poorly sorted and bedded, composed mostly of subrounded carbonate gravels from the Judea group (ca. 80%), subangular chert and few quartzite gravels, up to ca. 18 cm in size. The matrix is reddish sandy-clay-loam and contains disseminated carbonates as well as carbonate nodules, forming reddish-white mottling. Many gravels are covered by carbonate coating, up to 0.5 cm thick, while others are weathered. Some iron and manganese concretions are present at the top of the unit. Its contact with Unit 1 is abrupt.

In Area G, the unit is moderately sorted and most of the gravels are composed of chert. The matrix is grayish silty-clay, rich in large manganese and iron concretions that reach up to 2–3 cm in size. These concretions compose ca. 50% of the matrix. Manganese coating was identified on many of the larger gravels.

The matrix of Unit 2 is reddish, quartz-rich (58%), with significant amounts of aluminum and iron oxides and is poor in calcite (Fig 3) indicating a good drainage. The matrix of the gravel unit is probably derived from eroded Terra Rosa soils that covered the slopes of the drainage basin. These well-drained soils are composed of dust-related quartz, clay minerals and iron oxides.

**Unit 3** is carbonate-rich, pinkish silty-clay ca. 1.3 m thick, containing a few small chert and limestone gravel. It was identified only in the eastern section of Area A (Fig 2), where the entire sequence is markedly topographically higher in comparison to the other excavation areas. The

lower part of this unit has an angular blocky (ABK) structure, it is hard and compacted, and contains carbonate nodules 2–3 cm in size and carbonate bridges, as well as some manganese and iron concretions. It represents a K2.3/K3 calcic horizon of a well-developed calcic soil.

The upper part of Unit 3 has platy structure characteristic of K2.2 horizon of calcic soil. The sediments are disseminated with carbonate and nodules 2–3 cm in size. This upper part represents a well-developed calcic soil (calcrete) stage V–VI. Its contact with Unit 2 is clear, wavy.

Unit 3 is rich in calcite (75% on average) and poor in quartz (15% in average), aluminum oxide (Si/Al ratio–about 1.85) and iron oxide Table 2 and Fig 3). The accumulation of calcite indicates limited drainage and aerobic conditions.

**Unit 4** is a clast supported gravelly unit (ca. 70% gravel) 0.3–0.8 m thick, containing numerous flint artifacts. It consists of subangular-subrounded greyish-reddish chert gravel and few dolomites from the Judea Group. Gravel size range between 4 and 20 cm with a maximum of 25 cm. It is poorly bedded and moderately-well sorted. Large gravels are often covered with black manganese or yellow iron oxide coating. An occasional patina cover was observed.

The matrix is yellowish-brown-grayish silty-clay to clay, containing chert granules as well as manganese and iron oxide concretions up to 0.5 cm in size, which constitute ca. 25% of the matrix. These indicate gley and reduction conditions, suggesting hydromorphic soil that can be related to high groundwater and local wetland / marsh.

The upper part of the unit is finer in texture and composed of less gravel (20%) and more matrix, which is disseminated with carbonates, including large carbonate nodules (ca. 5 cm) and bridges. Large gravel often presents a second, carbonate coating, representing a K2 horizon of a calcic soil. Unit 4 is sealed by a calcrete stage III–IV.

Archaeological artifacts were found embedded mostly on the upper part of Unit 4, indicating that human activity at the site took place after the deposition of this unit, or in its final stages.

In Area E, the unit is only 10–15 cm thick and composed of relatively few fine chert gravels. Its lower boundary with the underlying Units 1, 2 and 3 is abrupt, indicating alluvial truncation, (Fig 2). The matrix of Unit 4, (Fig 2), which is usually rich in quartz (60% in average, Table 2) with moderate amounts of aluminium and iron oxides (Si/Al average ratio– 3.6), and poor in calcite. The gley, iron and manganese concretions and the greyish-yellow colors suggest reduction and hydromorphic conditions with presence of groundwater and anaerobic conditions. The unit is overtopped by a calcic soil, indicating limited drainage and aerobic conditions.

In Area G, Unit 4 is composed of five alternating gravelly and fine-grained sub-units with clear contacts (Fig 2):

Sub-units 4c, 4e are clast supported, poorly bedded, moderately-well sorted gravel units, containing ca. 60% gravel and numerous flint artifacts. The gravels are angular-subangular and mostly consists of chert up to 5 cm in size. The matrix is grayish-brown silty-clay/silty sand, rich in large black manganese and yellowish iron oxide concentrations 2–3 cm in size. Artifact density increases in the uppermost Sub-unit 4e. Four archaeological horizons were identified in the gravelly Sub-units of Area G, numbered (youngest to oldest) G1–G4. The oldest gravelly horizon was numbered G5, correlating to Sub-unit 4a. Lithic artifacts were scarce in that horizon.

Sub-units 4b, 4d are composed of massive brownish-grayish clay and sandy-clay-loam, respectively. Sediments are moderately hard and contain large amounts of iron and manganese concretions 2–3 cm in size and few chert and limestone gravels, up to 2 cm in size. These are characteristics of limited low-energy flows. Archaeological finds are scarce. These sub-units are particularly rich in aluminum and iron oxides.

**Unit 5** is a massive, compacted greyish-brown sandy clay layer, 0.4–1.6 m thick (Fig 2). The unit has a blocky structure, it contains few chert gravels and manganese concretions, as well as charred old roots. Large disorthic carbonate nodules with voids, up to 5–6 cm in size are present, more frequently at the base of this unit, and along joints as well as some iron and manganese concretions. Lower contact with Unit 4 is abrupt-to-clear wavy.

Unit 5 is rich in quartz (72% on average, Table 2) with moderate amounts of aluminium oxide (14% on average) and iron oxide (3.5% in average) and poor in calcite (Fig 3).

**Unit 6** is massive, compacted, greyish-brown clay unit, 0.6–1.3 m thick (Fig 2). Its structure is prismatic, with clear slickensides. Sediments contain large carbonate nodules (5–6 cm) and carbonate accumulations along slickenside plains, as well as few chert gravels, manganese concretions and roots. Its lower contact with Unit 5 is clear-gradual wavy.

**Unit 7** is a 1.0–1.8 m thick massive silty-clay grumusol. Its upper part, possibly A horizon, is moderate-slightly hard, with a blocky (BK) structure, and contains organic material. Its lower part, possibly B horizon, has a prismatic structure with slickensides and contains few chert gravels. Large carbonate nodules appear at the bottom of the unit. Lower contact with Unit 6 is gradual (Fig 2).

## Archaeological stratigraphy and spatial patterns

An archaeological horizon ca. 10–35 cm thick was exposed at the top of Unit 4 in all excavated localities (S1–S6 Figs). The extant of the excavated area as well as the depth and thickness of the archaeological find bearing unit in each of the localities are detailed in Table 3. Artifacts presented various degrees of abrasion, implying dynamic deposition environment and post-depositional processes. Nevertheless, for the most part, artifacts were found in a very good state of preservation, presenting sharp edges and prominent scar ridges. Heavily abraded artifacts, presenting dulled edges and often a thick coverage of patina, were also found in each of the excavated areas, albeit in small numbers.

The best state of preservation was noted at the northeastern part of the site, in Areas B, D, E (Figs 4 and S2, S4 and S5) and G (Figs 5 and S6), where artifacts mostly presented sharp edges and prominent scar ridges. Average layer thickness and find density, adding up to thousands of flint artifacts, are shown in Table 4. Artifacts in Areas B, D and G areas were mostly found in horizontal orientation and items smaller than 2 cm (chips) were abundant. The presence of several flat limestone slabs, possibly representing anvils or work surfaces, further implies that artifacts in these areas were subjected to little post-depositional movement.

**Table 3. Excavation details per locality.** Lithic assemblages include all artifacts collected, including from sieving using 5 mm mesh.

| Locality | | Excavated area (m²) | Average thickness of the archaeological horizon (m) | Estimated volume of excavated sediments (m³) | Size of lithic assemblage (N) | Average density for cubic meter (N) |
|---|---|---|---|---|---|---|
| **Area A** | | 11 | 0.30 | 3.3 | 12,011 | 3640 |
| **Area B** | | 6 | 0.35 | 2.1 | 16,413 | 7816 |
| **Area C** | | 13 | 0.20 | 2.6 | 6,840 | 2631 |
| **Area D** | | 16 | 0.30 | 4.8 | 24,136 | 5028 |
| **Area E** | | 12 | 0.08 | 0.96 | 5,247 | 5466 |
| **Area G** | G1 | 27 | 0.20 | 5.4 | N/A | - |
| | G2 | 16 | 0.15 | 2.4 | N/A | - |
| | G3 | 6 | 0.06 | 0.36 | N/A | - |
| | G4 | 4 | 0.08 | 0.32 | N/A | - |
| | G5 | 2 | 0.05 | 0.1 | N/A | - |

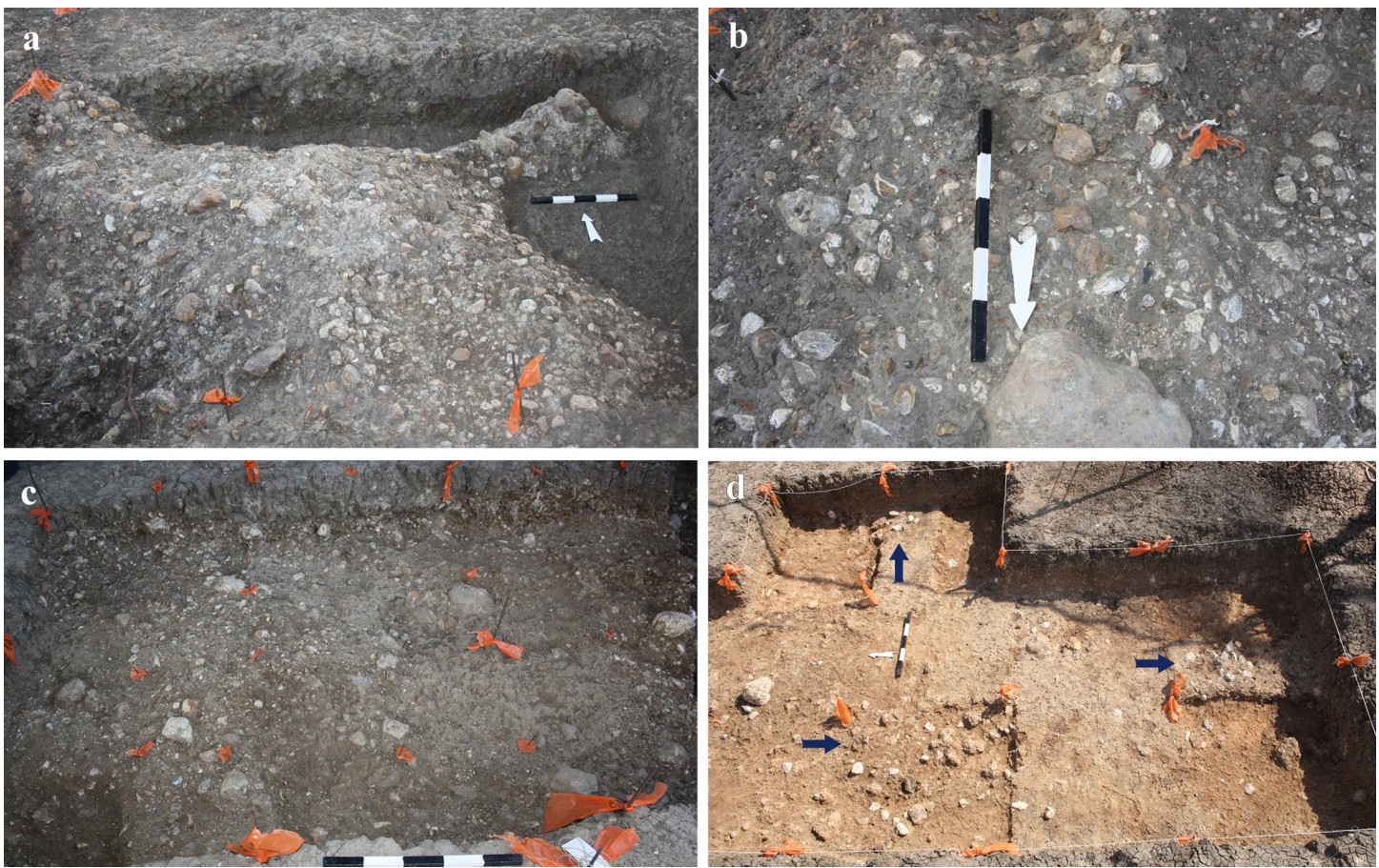

**Fig 4. The exposed archaeological horizons as displayed in the different localities.** a) Area A; b) Area D; c) Area B; d) Area E (arrows mark the location of a find cluster).

In Area G a sequence of four archaeological horizons was exposed (Figs 5B and S6), correlating to the gravelly Sub-units 4e, 4c and 4a (Fig 2). The uppermost horizons G1 and G2 were both identified in Sub-unit 4e, whereas horizons G3 and G4 correlated to Sub-units 4c and 4a, respectively. While the analysis of the finds is still in progress, a trend in the increase in find density between the lowermost G4 and the uppermost horizons G1–G2 was observed during fieldwork.

In Area D, post-depositional movement of artifacts was implied by their orientation, which often presented an oblique angle, and by the morphology of Unit 4. Faunal remains, only found in this area of the site, show evidence of water activity, manifested as edge rounding and polishing. At the western part of Area D, Unit 4 was composed of a series of moderate rises and depressions, indicating a dynamic fluvial environment at some point during its deposition, and possibly after the human activity in that locality. In the eastern part of Area D, however, the state of preservation was better, resembling that observed in Areas B and G.

In the southern part of the site (Areas A and C) the effects of post-depositional processes seemed to be more pronounced, indicated by the sharply rising and sloping morphology of Unit 4. Many of the artifacts in these localities presented an oblique orientation, following the sloping inclination of Unit 4. The abundance of highly abraded artifacts seemed to be higher in these areas, based on field observations.

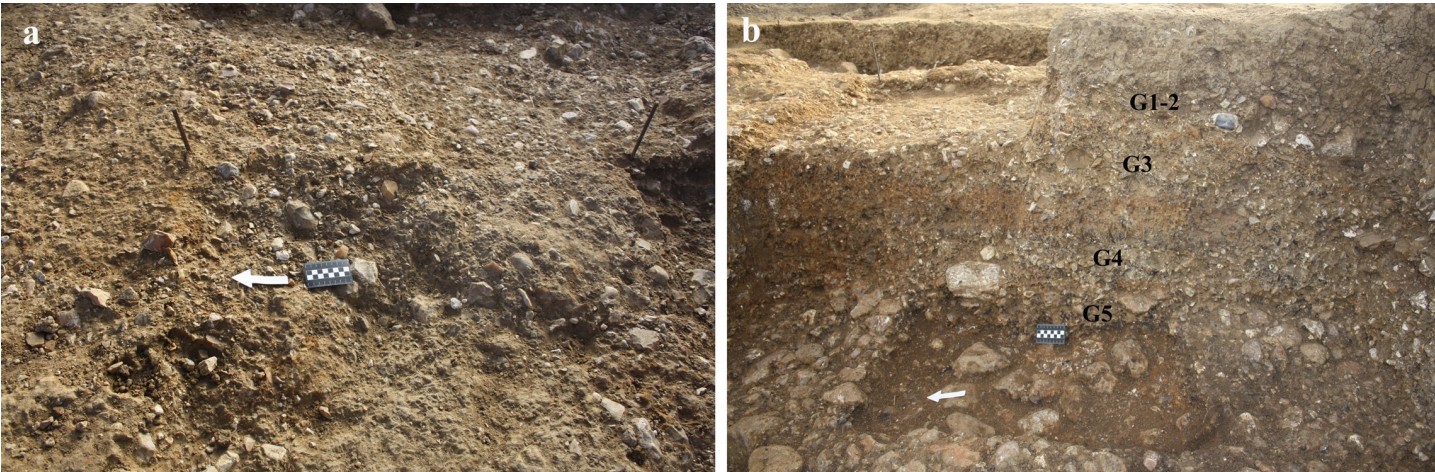

**Fig 5.** Area G: Horizon G2 (a) and the inner stratigraphy of Unit 4 exposed in that locality (b).

Finally, the westernmost Area E represented a somewhat unique phenomenon in comparison to the other excavated localities. Here, Unit 4 was very thin (less than 10 cm), and the density of gravel very low, possibly indicating the margins of the fluvial environ in which Areas A-D and G were deposited. Artifacts were found in three main clusters, each of them ca. 0.5 m in diameter, embedded on top of the Hamra soil of Unit 1 (Fig 4D). While find density was extremely low in comparison to the other excavated areas, artifacts were found in an excellent state of preservation, implying they were not subjected to post depositional movement.

**Table 4. Paleomagnetic interpretations of the samples from Jaljulia (Area B).**

| Horizon | Unit sampled | Sample | Treatment | Polarity | Interpretation |
|---|---|---|---|---|---|
| GAL 1 | 1 (Bottom) | GAL 1a | AF | N | N |
| | | GAL 1b | AF | N | |
| | | GAL 1c | AF | ? | |
| | | GAL 1d | T | ? | |
| GAL 2 | 1 (Middle) | GAL 2a | AF | N | N |
| | | GAL 2b | AF | N | |
| | | GAL 2c | AF | N | |
| | | GAL 2d | T | N | |
| GAL 3 | 1 (Top) | GAL 3a | AF | N | N |
| | | GAL 3b | AF | N | |
| | | GAL 3c | AF | N | |
| | | GAL 3d | T | N | |
| GAL 4 | 5 (Bottom) | GAL 4a | AF | N | N |
| | | GAL 4b | AF | N | |
| | | GAL 4c | AF | N | |
| | | GAL 4d | AF | N | |
| GAL 5 | 5 (Middle) | GAL 5a | AF | N | N |
| | | GAL 5b | AF | N | |
| | | GAL 5c | AF | N | |
| | | GAL 5d | AF | N | |

## Age estimation

**Paleo-magnetic stratigraphy.** Paleomagnetic interpretations are presented in Table 4 (Orthogonal comparative display of representative results is presented in S7 Fig). All experiments yielded clear results except for two samples from GAL1, and data quality is very good. All five tested horizons presented Normal polarity indicating that their deposition, including that of the artifact-bearing Unit 4, took place during the Brunhes Chron, and are therefore younger than ~780 ky, i.e., Middle Pleistocene.

**Luminescence dating.** The results of the luminescence analyses and proposed absolute ages for the sampled units are presented in Table 5, which also details the dose rates and equivalent doses for each sample. The dose rates vary from 1.1 to 2.8 Gy/ka, reflecting the varying proportions of fine sediments, gravels and secondary carbonates, and post-depositional processes.

The average De and errors were calculated using the central age model (CAM) after removing distinct outliers. OD–Overdispersion. No. aliquots–the number of aliquots used for the average De out of those measured.

Fig 6 displays representative luminescence signals and dose response curves for the TT-OSL and pIR-IR$_{290}$ signals, as well as a distribution of the aliquots measured for one sample. Both signals are bright and decay to background levels. Measured g-values (rate of fading) average 2.1%/decade. The De values of the pIR-IR$_{250}$ are ~ 80% of those measured for the pIR-IR$_{290}$, probably resulting from the higher fading rates of the pIR-IR$_{250}$ signal.

Overall, the pIR-IR$_{290}$ ages agree very well with the TT-OSL ages (Fig 7), increasing our confidence in the results. It appears that for the older samples (JAL 1 and JAL 7), the pIR-IR$_{290}$ ages are somewhat older than the TT-OSL ages. That could be explained by the thermal instability of the TT-OSL signal that causes some age underestimation at high De values.

Three samples were dated from Unit 5, the uppermost unit that was sampled. The quartz TT-OSL De values have a narrow range, from 375 to 403 Gy, however, due to highly varying

**Table 5. Field data and luminescence ages, following the measurement protocols detailed in 2.**

| LabCode | Unit | Area | Depth (m) | Dose rate (Gy/ka) | No. aliquots | OD (%) | De (Gy) | Age (ka) |
|---|---|---|---|---|---|---|---|---|
| JAL-10 QZKF | **5** | B | 3.0 | 1.59±0.06 | 11/12 | 26 | 378±11 | **239±11** |
| | | | | 2.07±0.10 | 10/10 | 8 | 478±14 | **231±13** |
| JAL-3 QZ | **5** | D | 4.1 | 1.09±0.04 | 8/8 | 22 | 375±30 | **346±31** |
| KF | | | | 1.53±0.08 | 6/6 | 15 | 512±33 | **335±28** |
| JAL-6 QZ | **5** | C | 3.1 | 1.35±0.05 | 8/8 | 8 | 403±12 | **299±14** |
| KF | | | | 1.81±0.09 | 6/6 | 10 | 515±24 | **284±19** |
| JAL-9 QZ | **4** | B | 3.2 | 1.27±0.06 | 8/8 | 13 | 400±19 | **316±20** |
| JAL-8 QZ | **4** | B | 3.4 | 1.45±0.06 | 8/8 | 14 | 532±27 | **367±24** |
| JAL-5 QZ | **4** | C | 3.5 | 0.83±0.04 | 8/8 | 1 | 357±18 | 428±25 |
| JAL-4 QZ | **4** | D | 4.4 | 0.69±0.03 | 8/8 | 23 | 353±28 | 513±46 |
| JAL-20 QZ | **4b** | G | 2.1 | 1.57±0.06 | 8/8 | 6 | 479±12 | **305±14** |
| KF | | | | 2.07±0.10 | 10/10 | 0 | 641±14 | **310±16** |
| JAL-21 QZ | **4d** | G | 2.0 | 2.33±0.10 | 7/8 | 15 | 456±12 | **195±10** |
| KF | | | | 2.92±0.14 | 9/10 | 15 | 583±16 | **200±11** |
| JAL-7 QZ | **1** | B | 5.0 | 2.23±0.08 | 7/7 | 12 | 664±30 | **298±17** |
| KF | | | | 2.77±0.12 | 10/10 | 8 | 878±29 | **317±17** |
| JAL-1 QZ | **1** | E | 5.0 | 1.16±0.05 | 8/8 | 5 | 446±10 | **385±17** |
| KF | | | | 1.61±0.08 | 10/10 | 13 | 705±33 | **438±31** |

QZ–quartz, measured using the thermally transferred OSL (TT-OSL) signal.

KF–alkali feldspar, measured using the post-IR IR at 290˚C (pIR-IR$_{290}$) signal.

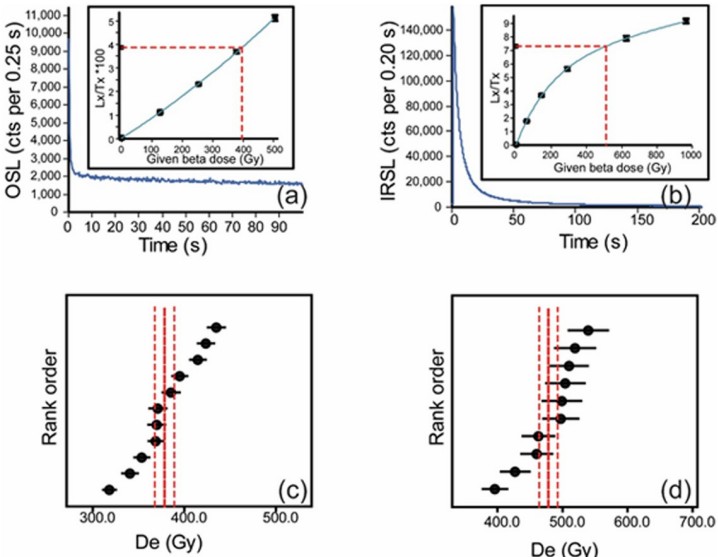

**Fig 6. Luminescence results for sample JAL-10.** Aliquot size, stimulation source, fits and integration channels are listed in Table 2. (a) Natural TT-OSL signal (in blue) for one large aliquot (~1500 grains) of quartz grains, displaying the rapid decay of the signal within a few seconds, and the slow background decay. Inset: Dose response curve for the same aliquot. De = 395±8 Gy. The recycling ratio (RR; the repeated measurement point at 120 Gy) = 1.07 and recuperation = 0.95%. (b) Natural pIR-IR$_{290}$ signal (in blue) for one small aliquot (~50 grains) of alkali-feldspar grains. Inset: Dose response curve for the same aliquot. De = 511±29 Gy. RR = 0.99 and recuperation = 0.4%. De is not sensitive to the selected channels. (c) Rank order plot for TT-OSL measurements, showing all 11 accepted aliquots. Average De, calculated using the central age model (CAM), is 378±11 Gy, and over-dispersion (OD) = 9%. (d) Rank order plot for pIR-IR$_{290}$ measurements, showing all 10 aliquots. Average De, calculated using CAM, is 478±14 Gy and OD = 8%.

dose rates, the ages range from 239±11 ka in Area B to 364±31 ka in Area D, averaging 261±33 ka. The pIR-IR$_{290}$ ages follow a similar trend, with a narrow range of De values but ages ranging from 231±13 ka in Area B to 335±28 ka in Area D.

Six samples were collected from Unit 4 (Table 5), of which three were collected from within the archaeological horizon itself (JAL 4–5, 9), one from the underlying gravels in Area B (JAL

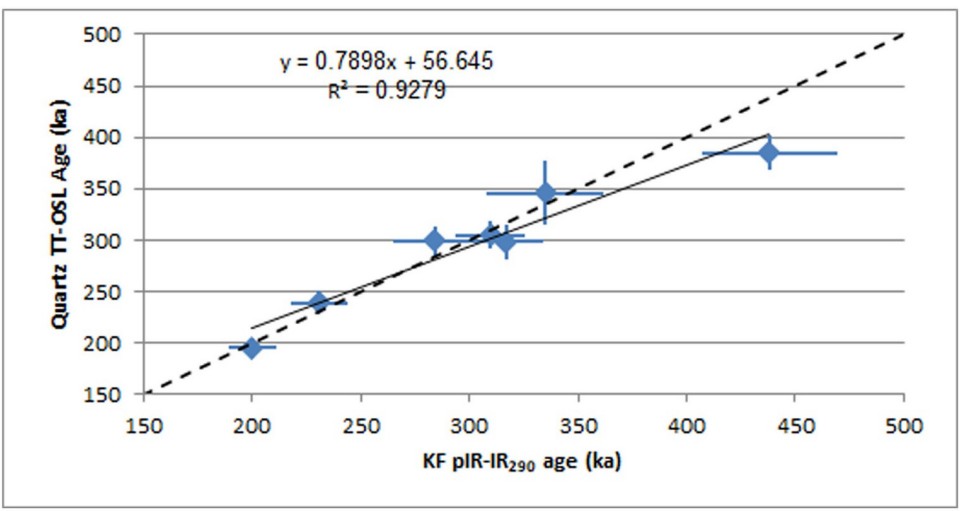

**Fig 7. A comparison of ages obtained by TT-OSL on quartz and pIR-IR290 on KF.**

8) and two from the silty Sub-units 4d and 4b in Area G (JAL 21 and JAL 20, respectively; Fig 2). All six samples were measured by TT-OSL, but only two samples were also measured by pIR-IR$_{290}$ (JAL 20–21). The TT-OSL De values range from 357 to 532 Gy, and the pIR-IR290 De values from 583 to 641 Gy. Dose rates vary greatly, and as a result also the ages, ranging from 195±10 ka in Area G to 513±46 ka in Area D (Table 5).

The TT-OSL age of the lowermost sample, JAL 8, is 367±24 ka; this age is most likely underestimated as the De is greater than 500 Gy, where the thermal instability of the signals starts to take affect [91]. The TT-OSL De values of the overlying two samples, 4 and 5, of ~355 Gy, are well below 500 Gy, thus, the ages obtained for these samples, 428±25 ka and 513±46 ka, can provide a good estimate of the age of this unit. The third sample from this unit, JAL 9, from Area B and slightly higher in the section, gave a TT-OSL age of 316±20 ka. The uppermost two samples, JAL 20 and JAL 21, from Area G, gave an age of ~310 ka for Unit 4b and ~200 ka for Unit 4d.

Two samples were collected from the lowermost Unit 1, a Hamra soil with well-developed calcrete (JAL 1, 7), giving ages of 300–430 ka. The high TT-OSL De value of JAL-7, 664 Gy, suggests underestimation, and may provide only a minimum age for the unit. The pIR-IR$_{290}$ De value for that sample, 705 Gy, is not yet in the range of saturation for the signal. However, the age of 438±31 ka seems too young considering the depositional history of the site and is probably also an underestimation, which could be explained by a change of dose rates over time.

**ESR chronology.** ESR chronology results are displayed in Table 6 (ages are given with ±1σ). Growth curves are shown in Fig 8. The ages determined by the Al and Ti-Li centers agree for all the measured samples, except for JAL6, for which Al and Ti-Li ages differ by more than 100 ka. All samples from Unit 4 fall within the range of 462–330 ka, similar to the age range provided by luminescence methods. Al and Ti-Li ages are coherent for all samples but JAL9, for which a Ti-Li age calculation was not possible due to a wide dispersal of the measurement points on the growth curve and to a large error range.

The silty clays of Unit 5 were dated in Areas C and D (JAL 6 and JAL 3, respectively). ESR yielded an age range of 378–220 ka while luminescence provided a more limited age range for these two samples, roughly 350–300 ka. JAL6 had a lower De using Al center with a large error resulting from the difficulties due to the variation of signal intensity for each measurement in relation with the heterogeneity of the sample.

## The lithic industries

The analysis of lithic artifacts was conducted following the methodology detailed in previous studies by Rosenberg-Yefet et al. [42, 92]. The lithic assemblages from all excavated areas show classic affinities of the Late Acheulian of the Levant, containing handaxes, bifaces and rich

**Table 6. ESR age estimation results.**

| Locality | Unit | Samples | Z (cm) | dBl (%) | $D_a$ (mGy/a) | $D_b$ (mGy/a) | $D_{cos}$ (mGy/a) | Dose rate (mGy/a) | $D_e$ (Gy) | $r^2$ | Age (ka) |
|---|---|---|---|---|---|---|---|---|---|---|---|
| B | 4 | JAL9 Al | 300–320 | 53 | 60 ± 1 | 588 ± 11 | 118 ± 6 | 1292 ± 15 | 425 ± 33 | 0.995 | 330 ± 30 |
| | | JAL9 Ti-Li | | 100 | 60 ± 1 | 588 ± 11 | 118 ± 6 | 1292 ± 15 | *789 ±130* | | NC |
| C | 4 | JAL5 Al | 350 | 53 | 50 ± 1 | 483 ± 15 | 111 ± 6 | 1077 ± 21 | 478 ± 34 | 0.995 | 444 ± 30 |
| | | JAL5 Ti-Li | | 100 | 50 ± 1 | 483 ± 15 | 111 ± 6 | 1077 ± 21 | 480 ± 56 | 0.976 | 446 ± 30 |
| | 5 | JAL6 Al | 320 | 55 | 51 ± 2 | 576 ± 19 | 116 ± 6 | 1253 ± 25 | 276 ± 71 | 0.982 | 220 ± 40 |
| | | JAL6 Ti-Li | | 100 | 51 ± 2 | 576 ± 19 | 116 ± 6 | 1253 ± 25 | 420 ± 76 | 0.961 | 335 ± 50 |
| D | 5 | JAL3 Al | 410 | 49 | 45 ± 1 | 523 ± 15 | 101 ± 5 | 1133 ± 20 | 428 ± 78 | 0.961 | 378 ± 70 |
| | | JAL3 Ti-Li | | 100 | 45 ± 1 | 523 ± 15 | 101 ± 5 | 1133 ± 20 | 363 ± 33 | 0.988 | 320 ± 30 |
| | 4 | JAL4 Al | 430 | 52 | 35 ± 1 | 378 ± 13 | 98 ± 5 | 840 ± 17 | 388 ± 17 | 0.998 | 462 ± 60 |
| | | JAL4 Ti-Li | | 100 | 35 ± 1 | 378 ± 13 | 98 ± 5 | 840 ± 17 | 350 ± 32 | 0.988 | 417 ± 50 |

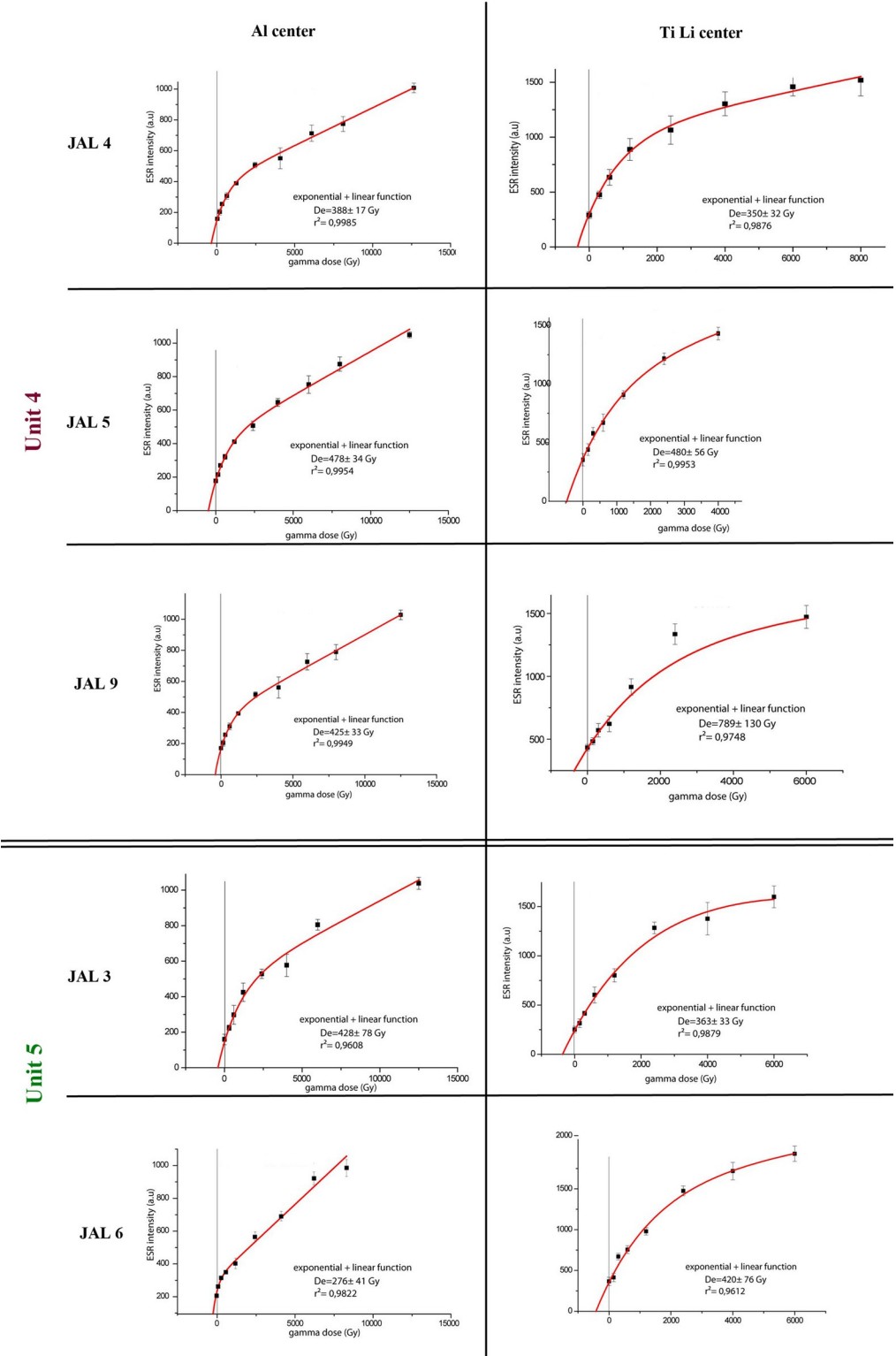

**Fig 8. Growth curves of the ESR samples.** Al centers are presented on the left. Ti Li centers are on the right.

flake industries (including prepared core technologies and cores on flakes). While full comparative analyses of the assemblages are beyond the scope of this paper, we present here an overview of the lithics from one locality: Area B, dated to ca. 367–316 ka.

The assemblage includes a total of 16435 flint items, of which ca. 56.3% are debris (Table 7). Shaped items are relatively abundant (ca. 12.28% of the assemblage), including notably high frequencies of bifaces (6.2% of the assemblage; n = 127).

Modified base flake are flakes which had their base modified prior to detachment; lip flakes are flakes having a bulb in the shape of a lip, usually associated with handaxe-thinning; the full definition of prepared cores appears later in the text; recycled products are small flakes detached from cores on flakes and are characterized by two flat and smooth faces (ventral-like) and in some cases also a straight, thin profile; special waste items are spalls detached from tools during modification, reshaping or resharpening (e.g. bifacial thinning flakes, burin spalls, scraper rejuvenation flakes); Micro-flakes are complete flakes with a bulb of percussion, which are smaller than 1.5 cm.

The vast majority of cores (more than 99%; Table 8) reflect the production of items bearing flake proportions. Among the cores, the production of relatively large and medium flakes from one-, two- or multiple-striking platform is the most abundant (ca. 73% of the cores; cores on flake excluded).

The production of small sharp flakes by means of lithic recycling (e.g., [93]) is shown by ca. 23% of the total amount of cores in the assemblage (cores and cores on flakes). Notably, the small flakes that were produced from these cores comprise ca. 3% of the debitage (n = 203, 'recycling products' in Table 7). Use wear analyses conducted on similar small flakes from the Late Acheulian site of Revadim indicated their use in the conduct of tasks demanding high precision during butchery activities [94]. Prepared cores (Fig 9), used for the production of predetermined flakes, constitute ca. 16.7% of the cores in the assemblage (cores on flakes excluded). Items were ascribed to this category in the spirit of the definition provided by

**Table 7. General breakdown of the flint assemblage from Jaljulia, Area B.**

| Area B | N | % of débitage and shaped items | % of total assemblage |
|---|---|---|---|
| Primary element flake (PE flake) | 1055 | 15 | 6.4 |
| Primary element blade (PE blade) | 44 | 1 | 0.3 |
| Non-modified base flake | 1320 | 18.8 | 8 |
| Modified base flake | 724 | 10.3 | 4.4 |
| Blade | 38 | 0.5 | 0.2 |
| Lip flake | 127 | 1.8 | 0.8 |
| Prepared core flake | 94 | 1.3 | 0.6 |
| Core trimming element (CTE) | 319 | 4.5 | 1.9 |
| Core | 629 | 9 | 3.8 |
| Core on flake | 195 | 2.8 | 1.2 |
| Recycled products | 206 | 2.9 | 1.3 |
| Shaped items | 2032 | 29 | 12.4 |
| Special waste | 229 | 3.2 | 1.4 |
| **Total débitage and shaped items** | **7012** | **100.00** | **42.7** |
| Chunk flake | 3735 | - | 22.8 |
| Chunk | 3094 | - | 18.9 |
| Chip | 1820 | - | 11.1 |
| Micro flake | 745 | - | 4.5 |
| Raw material | 7 | - | 0.04 |
| **Total** | **16413** | **-** | **100.00** |

Debénath and Dibble [95]: *"a number of technologies. . . in which the core was intentionally shaped or prepared in such a way as to predetermine the shapes of flakes taken from it"*. Here, we apply a technological subdivision into three categories:

1. Proto-Levallois cores (Fig 9: 1, 2; n = 49; ca. 47% of the 'prepared cores' category; ca. 6% of total cores) are defined, based on Picin [96] and on recent work by Rosenberg-Yefet and Barkai [35], and Rosenberg-Yefet et al. [92] as cores in which the volume is divided into two hierarchical surfaces, a striking platform and a flaking surface, with a partial or complete intersecting ridge separating the two surfaces. Striking platforms are usually prepared, and lateral and distal convexities of the flaking surface are configured by either *debordant* flakes or by preparational flaking of the core circumference. In many aspects, Proto-Levallois cores share conceptual similarities with the Middle Paleolithic Levallois method (e.g., [97]), albeit they do not fully accord with the complete set of the characteristics associated with the fully-fledged method.

2. Discoid cores (Fig 9: 3; n = 36; ca. 35% of the 'prepared cores' category; ca. 4% of total cores) were defined following the guidelines set by Terradas [98], describing cores divided into two platforms of equal volumes, which display alternating hierarchy or no hierarchy at all. These cores exhibit full\partial shaping of the circumference.

3. Prepared cores (general) (Fig 9: 4; n = 14; ca. 18% of the 'prepared cores' category; ca. 2% of total cores) are cores that clearly belong to the prepared cores group (volume is divided into two platforms, preparation on the circumference or scars indicating *debordant* removals) though do not meet the stricter definitions of the former two subcategories.

Evidence for the utilization of prepared cores technologies were also found in the presence of characteristic core trimming elements (CTE; ca. 19%), and of target end products within the debitage. The shaped items category is composed of retouched flakes and bifaces (Table 9).

Bifaces (Fig 10) are present in the assemblage in significant numbers (n = 127, 6.3% of all shaped items) in comparison to other Late Acheulian sites such as Revadim layer C5 (3%), and layer C3 (0.5%), and Berekhat Ram (1.98%) [9, 33, 35]. Bifaces were preliminary divided into

**Table 8. Techno-typological distribution of the cores in the assemblage (excluding cores on flakes).**

| Core type | N | % of category |
|---|---|---|
| **Flake cores** | **465** | **73.9** |
| Single striking platform | 241 | 51.8 |
| Two striking platforms | 141 | 30.3 |
| Multiple striking platforms | 83 | 17.8 |
| **Flake/Blade cores** | **1** | **0.2** |
| Multiple striking platforms | 1 | 100.0 |
| **Laminar cores** | **2** | **0.3** |
| Single striking platform | 2 | 100.0 |
| **Prepared cores** | **99** | **16.7** |
| Prepared cores (general) | 14 | 18.1 |
| Proto-Levallois cores | 49 | 46.7 |
| Discoid cores | 36 | 35.2 |
| **Core fragment** | **28** | **4.5** |
| **Tested core** | **24** | **3.8** |
| **Varia** | **4** | **0.6** |
| **Total** | **629** | **100.0** |

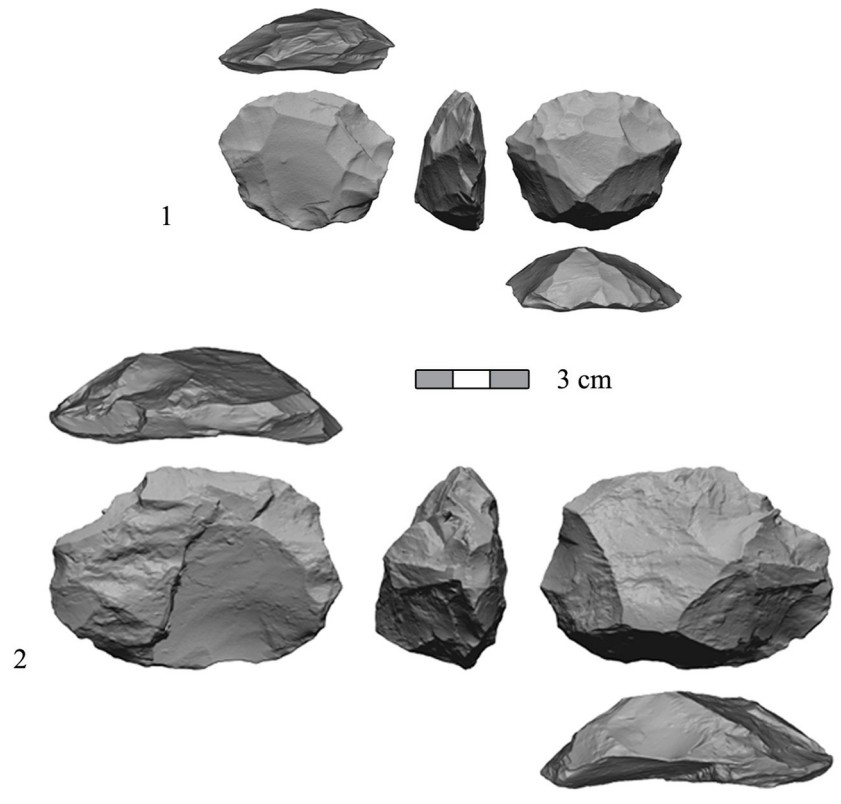

3 cm

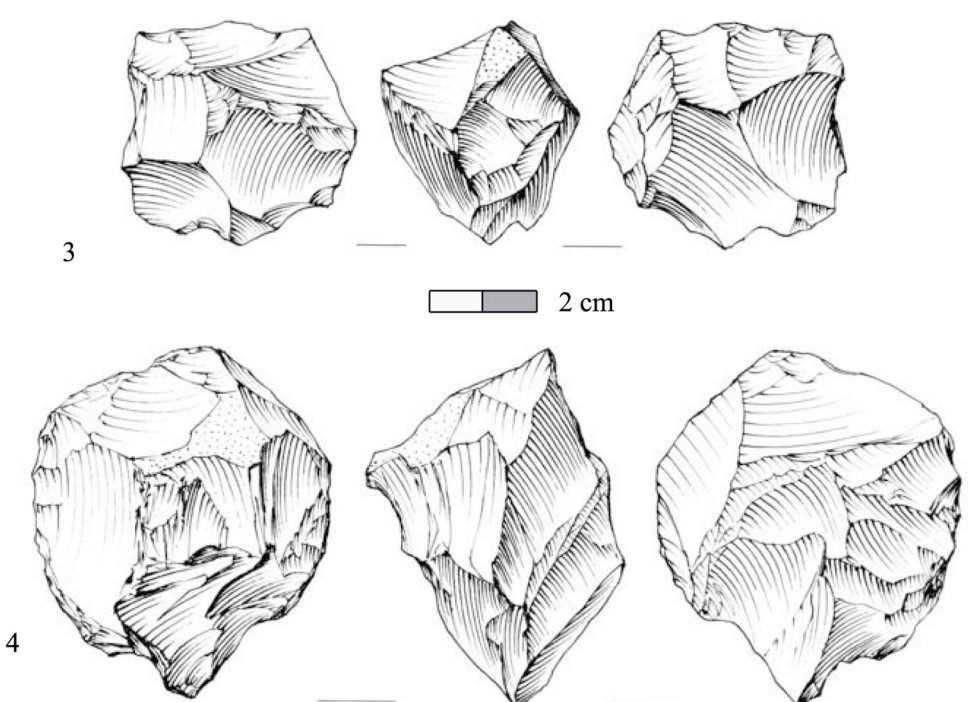

2 cm

**Fig 9. Prepared cores from the assemblage of Area B in Jaljulia.** 1–2) Proto Levallois; 3) Discoid; 4) Prepared core (general).

**Table 9. Shaped items typology and frequencies.**

| Shaped items | N | % of category |
|---|---|---|
| Partially retouched flakes | 756 | 37.2 |
| Partially retouched blades | 40 | 2.0 |
| Retouched items produced by the recycling of old flakes | 44 | 2.2 |
| Side scraper | 58 | 2.9 |
| Scraper fragment | 11 | 0.5 |
| End scraper | 17 | 0.8 |
| Notch | 164 | 8.1 |
| Truncation | 22 | 1.1 |
| Awl-Borer | 27 | 1.3 |
| Denticulated | 52 | 2.6 |
| Chopper | 12 | 0.6 |
| Biface | 127 | 6.3 |
| Varia | 257 | 12.6 |
| Retouched Fragments | 445 | 21.9 |
| **Total** | **2032** | **100.0** |

subgroups: general bifaces (ca. 10%); roughouts (ca. 9%); handaxes (ca. 66%; Fig 10: 3, 4); handaxe transformed into cores (ca. 4%) and handaxes with preferential flake scars (ca. 11%; Fig 10; for a detailed description of these handaxes see [42]). A more detailed analysis of the biface assemblage is currently in progress. The presence of bifacial debitage, such as thinning flakes (Fig 11: 1–5) and flakes removing the bifacial ridge (Fig 11: 6) within the debitage (the 'special waste' category) indicates that some stages of biface production and maintenance were conducted on site (the challenging distinguish between handaxe-thinning flakes and flakes detached from prepared cores was conducted based on bulb characteristics, flake profile and the general morphology of the dorsal scar pattern; see [35] for a more detailed treatment of this issue regarding late Acheulian Revadim). A preliminary use-wear study of a sample of bifaces from Area B revealed their use in activities mostly related to breaking bone [99]. Among the retouched tools, most dominant are partially retouched flakes, constituting ca. 37% of the total items in the category. Other types, such as notches (ca. 8%), side scrapers (Fig 11:10; ca. 3%) and denticulated (Fig 11:6–9; ca. 3%) appear in much lower frequencies. Shaped items made on blanks with laminar proportions are rare (ca. 2%).

## Faunal remains

Area D represented the only locality in which faunal remains were preserved (N = 38 bones). Of these, 12 fragments, representing a total of three skeletal elements, could be identified to body part, species or body size class: four fragments of a cervical vertebra of a large bovid, probably aurochs, *Bos primigenius*; three fragments of a large mammal rib, also probably of aurochs; and five fragments of an aurochs left astragalus (Fig 12). Unfortunately, no measurements could be taken on these remains and morphological features were partial. Consequently, the tentative attribution of the remains to aurochs, rather than bison, is based on biochronology. This, since aurochs (rather than bison) were a commonly exploited species in Late Acheulian sites in the Southern Levant [23].

For the reconstruction of the taphonomic history of the fauna, remains were examined under magnification (×10–×20) for evidence of biotic and abiotic surface modifications [100]. No anthropogenic (butchery or percussion marks) or other signatures of biotic damage (animal gnawing) were identified. However, all bones exhibit edge rounding and surface polishing that

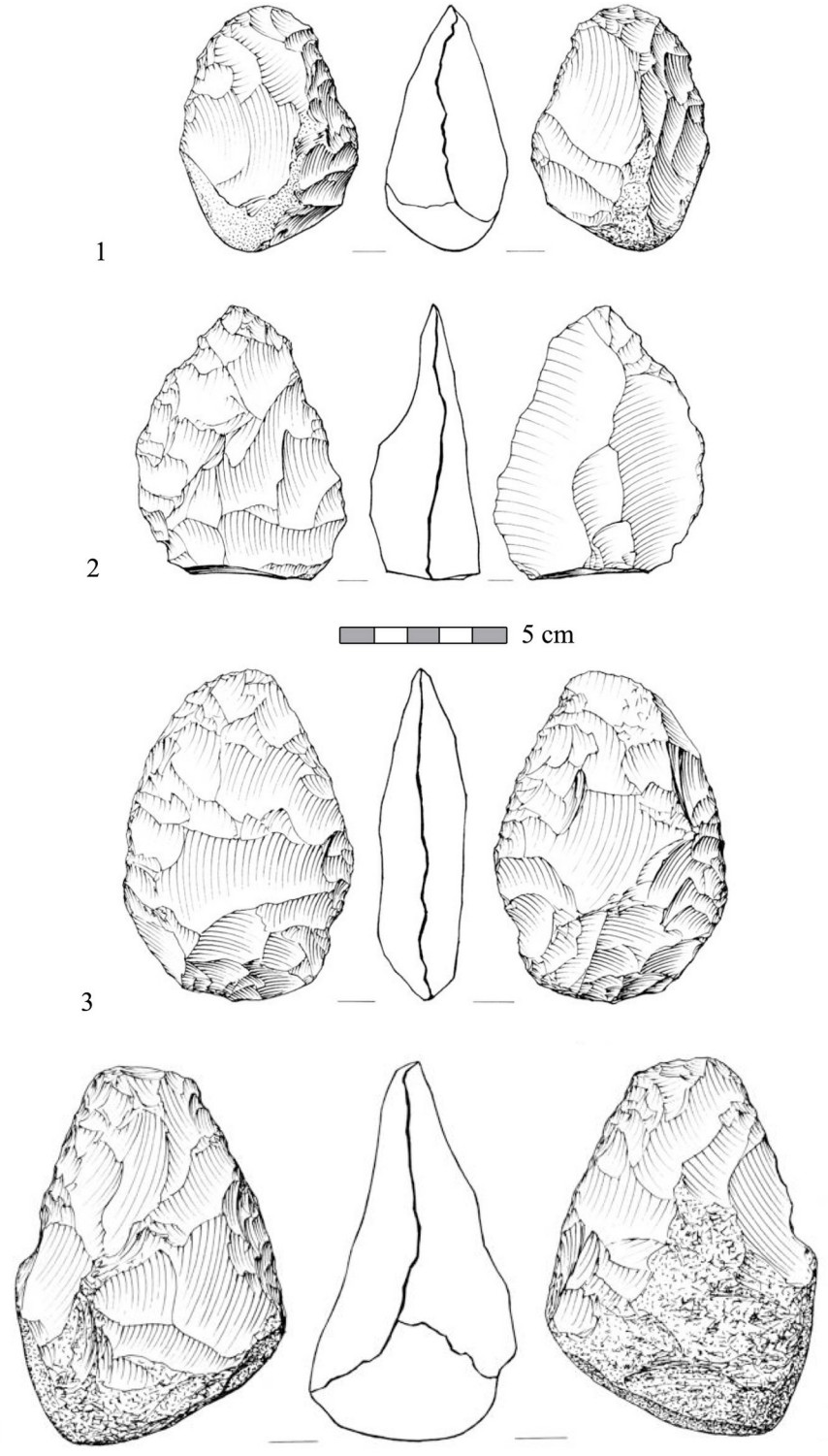

**Fig 10. Typical bifacials from the assemblage of Area B in Jaljulia.** 1, 3–4) Handaxes. 2) A handaxe with preferential flake scars.

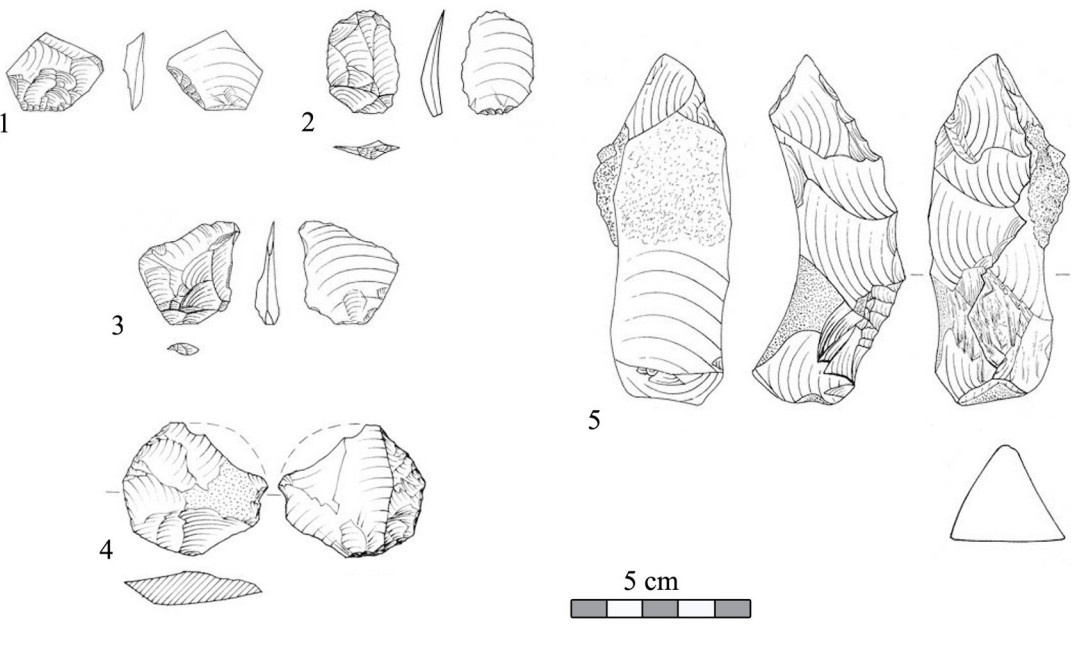

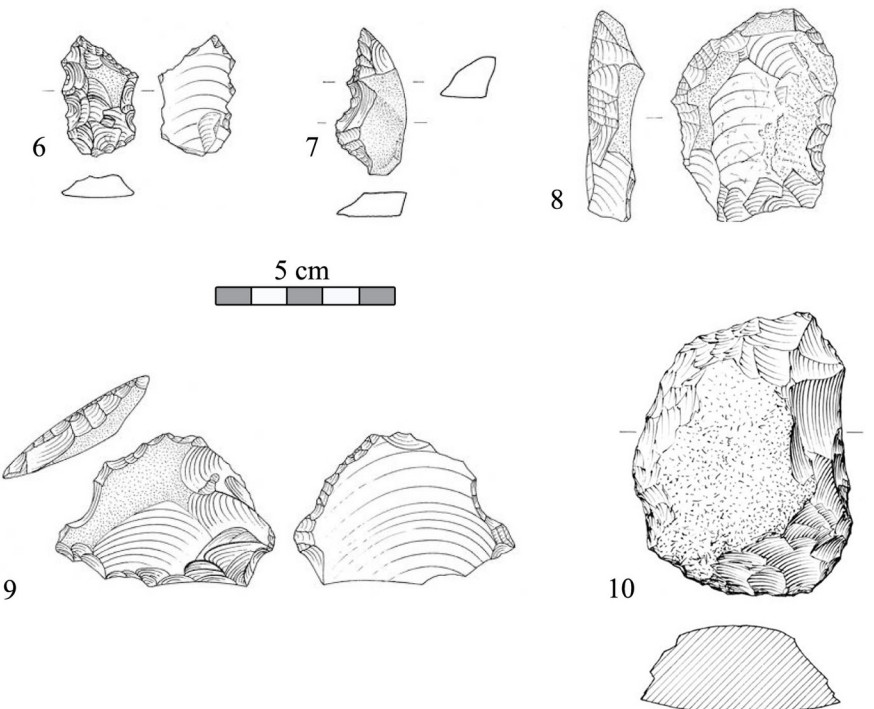

**Fig 11. Special spalls and shaped items from the assemblage of Area B in Jaljulia.** 1–5) Bifacial shaping spalls. 6–9) Denticulated flakes. 10) Scraper.

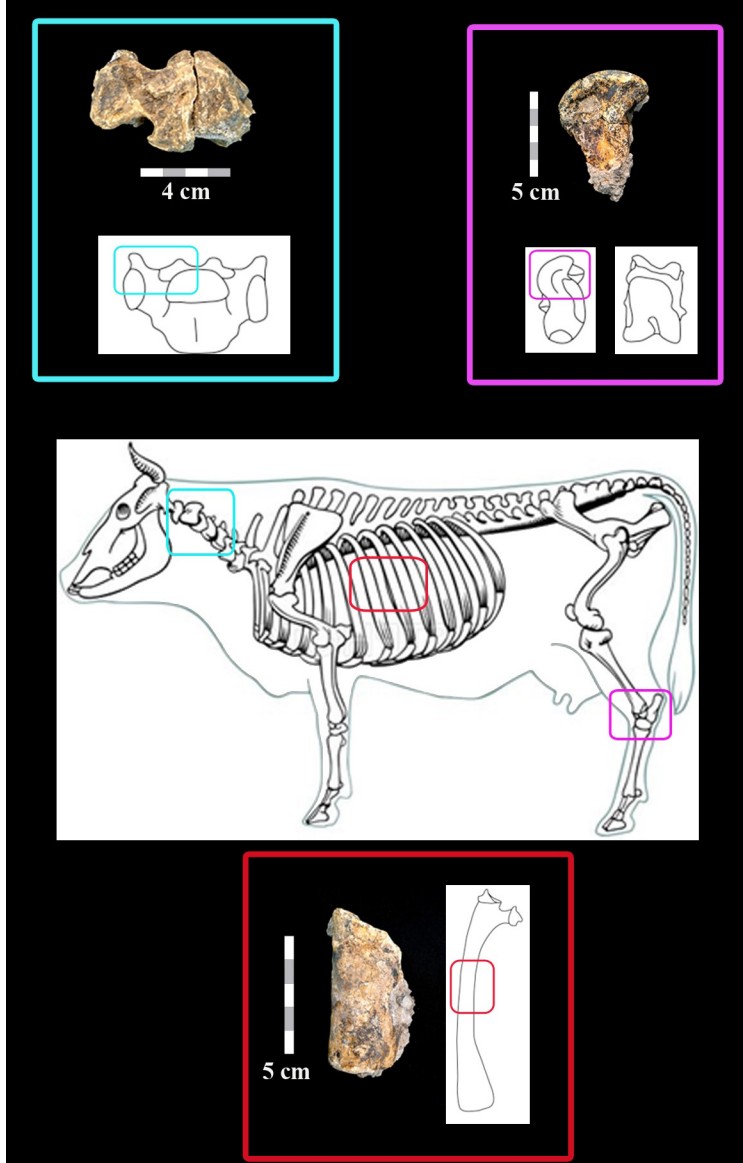

**Fig 12. Diagnostic bone fragments found in Area D.** All are from large bovids, probably aurochs.

is typical of abrasion, probably from water transport in coarse gravels [101] as occur in the sedimentary matrix of the site. Consequently, damage generated by surface exposure (weathering), breakage and even biotic modifications, has been obliterated by the abrasion. Additional indication of dynamic and strong hydraulic activity in this area is the undulating topography of the gravel deposit and the orientation of the lithic artefacts (both discussed above).

Since hydraulic transportability of animal remains is associated with the shape, size and density of bones [100, 102 and references therein], the unidentified portion of the assemblage, comprising 26 fragments, was measured to assess size and shape, and the extent of cortical to trabecular bone (i.e., bone density) was assessed for each fragment. This assemblage was composed of 15 mixed compact and trabecular (spongy) bones, 9 fragments of compact bone, and only two trabecular bones. The predominance of items comprising compact bone is undoubtedly the outcome of its greater robusticity and hence survivorship in an active fluvial environment.

In terms of size, one compact bone fragment is much larger than the others (66 mm long, 32 mm wide and 24 mm cortical bone thickness) and may possibly represent a shaft fragment of a mega-mammal i.e., from a hippopotamus or elephant. All others fall in the large mammal category with average length 22.0 mm, breadth 11.2 mm and thickness 8.2 mm. The identified material was larger with average length 32.9 mm, breadth 30.5 mm and thickness 22.7 mm. In their flume experiment (with maximum flow velocity of 40 cm/s), Pante and Blumenschine [101] noted that there was a significant but inverse relationship between transportability i.e., numbers of bone transported, with cortical thickness, maximum fragment length and width factors of decreasing importance. For example, optimal transport occurred in midshaft specimens i.e., long bone cortical shafts, with cortical thickness of 2 mm, maximum length ≤20 mm, width ≤6–10 mm. While the lengths and breadths of the Jaljulia fragments fit the size range for near optimal hydraulic transport, the cortical thickness measures do not. Clearly the strength of the current at the site would have been factors influencing transportability.

As shown in Fig 13 and Table 10, the majority of fragments (73%) have a similar shape and fall within the category of 'compact-elongate', 'elongate', 'very elongate' i.e. relatively longer than wide or thick, as defined in Sneed and Folk [103]; 23% fall within the category of 'compact-bladed', 'bladed' and 'very bladed' (i.e. thicker and wider than the elongated group relative to length), while only 4% fall in the 'platy' category (i.e. very thin and narrow). It is possible that the dominance of the elongated shape is the end result of abrasion and rolling in the fluvial system.

In order to explore bone diagenesis at the site, elemental composition of four bones was examined—two cortical bone and two trabecular—using Scanning Electron Microscopy (SEM) and energy dispersive X-ray spectroscopy (EDS). The results, given in Fig 14, demonstrate that all bones were highly fossilized and have undergone marked diagenetic changes. The results further demonstrate that there are significant differences in composition between cortical and spongy bone, as well as between light-colored cortical bone and the dark areas that spot over half of the bones' surfaces which are Manganese stains [e.g., 104].

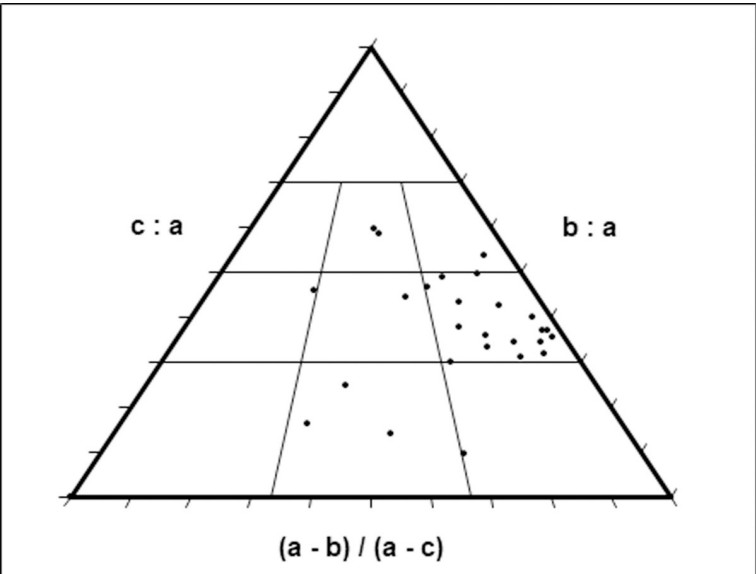

**Fig 13. Comparative bone dimensions plot.** Axes legends follow Sneed and Folk [103] where letters a, b and c represent the long, intermediate and short orthogonal axes of each bone fragment, respectively.

**Table 10. Numbers of bone fragments by composition and shape categories following Sneed and Folk [103].**

|  | Cortical | Trabecula | Cortical/Trabecula | Total |
|---|---|---|---|---|
| **Compact** | - | - | - | - |
| **Compact-Platy** | - | - | - | - |
| **Compact-Bladed** | - | 1 | 1 | 2 |
| **Compact-Elongate** | - | - | 1 | 1 |
| **Platy** | 1 | - | - | 1 |
| **Bladed** | 1 | - | - | 1 |
| **Elongate** | 3 | 1 | 12 | 16 |
| **Very Platy** | - | - | - | - |
| **Very Bladed** | 3 | - | - | 3 |
| **Very Elongate** | 1 | - | 1 | 2 |
| **Total** | 9 | 2 | 15 | 26 |

The localized nature of the faunal remains within the excavated area (Area D), the extensive abrasion leading to edge rounding and polish on all bones, the predominance of compact over spongy bone and the relatively elongated shape and similar size of all fragments, all support that the was exposed to hydraulic action. As such, they may not be directly associated with the archaeological artifacts but were introduced into this locality via fluvial action.

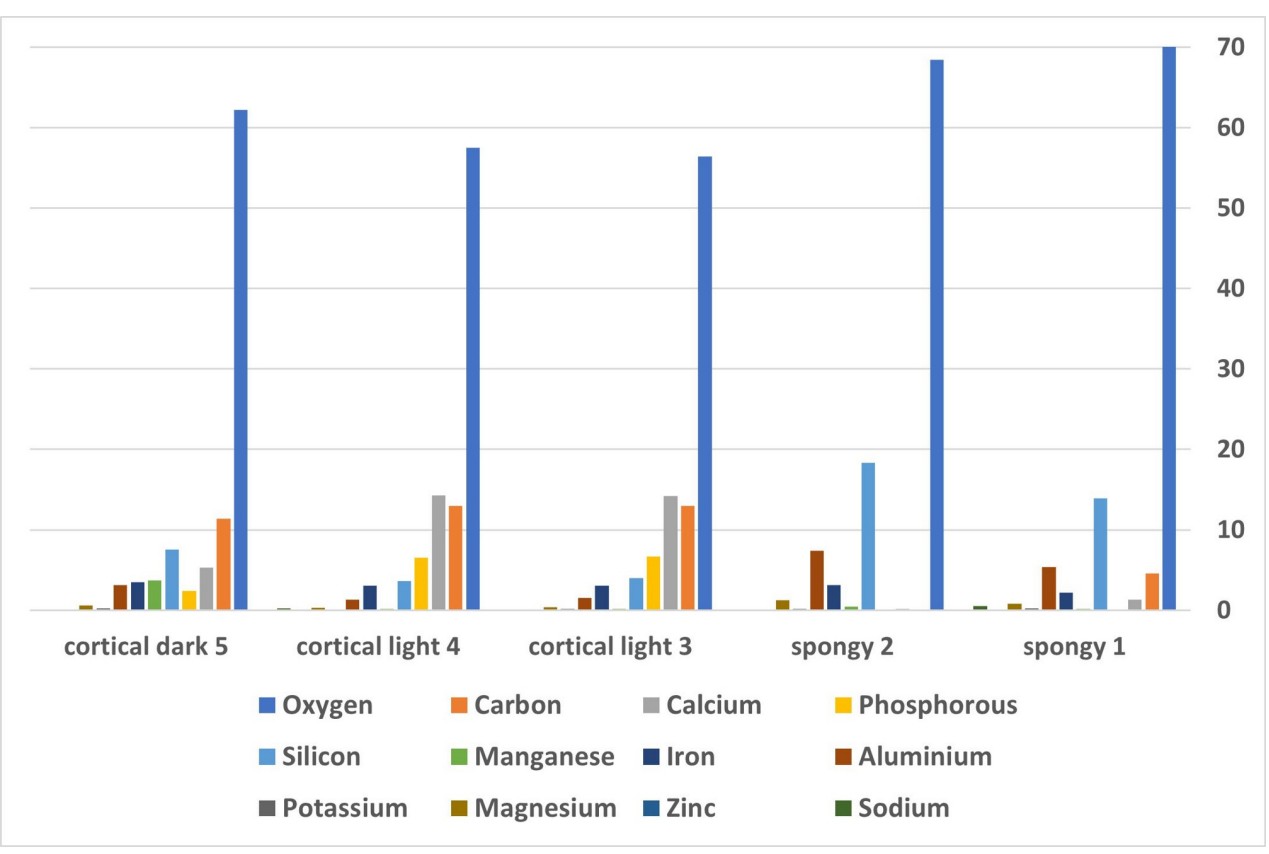

**Fig 14. Elemental composition of four bones given as atomic percentage.**

## Discussion

### Site formation and paleoenvironment settings for human activity

The study of the stratigraphy, geomorphology and sedimentology of the site allowed the reconstruction of the depositional history of the site and its immediate surroundings, shedding light on site formation and post-depositional processes, as well as on the characteristics of the paleoenvironment in which the human activity took place.

The sources of the sediments at the site are: (1) Coastal aeolian quartz sand from the west. (2) Aeolian dust composed of quartz, carbonates, silt and clays [105]. (3) Fluvial silt and clay eroded from the soils of the drainage basin. (4) Limestone and chert gravels sourced from the nearby drainage basin. (5) Aluminum oxides represent mostly clay minerals. Clay minerals and iron oxides, result from local chemical weathering in water-rich environments.

The main formation stages are described below from the oldest to the youngest (Figs 2 and 15):

1. Middle Pleistocene aeolian quartz sand of Unit 1 was deposited over the coastal plain and further inland, almost to the foothill of the Judea Mountains [63]. The unit is exposed at the village of Jaljulia, just west of the site, and mapped as Pleistocene/Holocene Red sand and loam (Fig 1C). In places, the sand lithified into aeolianite ('Kurkar'), forming parallel, longitudinal (north south), low ridges [106–111] mapped as Cross-bedded, Pleistocene Calcareous sandstone (Fig 1C). The normal magnetic polarity of Unit 1 indicates that the deposition of these sediments was during the Brunhes Chron, and that they are younger than ~780 ky. Using TT-OSL, Harel et al. [111], documented and dated dune deposits 7 km east of the present coastline to 446±41 ky, showing that the age of the dunes increases with the distance from the shoreline. Nevertheless, they estimated that there may be buried older aeolianites to the east. The site of Jaljulia is located ca. 14 km from the shoreline and therefore the age of the sand can be assumed to be even older. Therefore, the range of luminescence ages for Unit 1 in Jaljulia (ca. 438–298 ka) is young and should be treated as minimum ages.

2. At a later phase, and probably under slightly wetter conditions, these sediments were exposed, and sandy-loam Hamra soil developed [112–114]. The Hamra soils are also a part of the sandy phase mapped as Pleistocene/Holocene Red sand and loam (Fig 1C). The presence of reddish iron oxides at that stage indicate aerobic conditions, good drainage, and intensive chemical weathering into clay minerals, dominated by illite-smectite and minor kaolinite [115]. Sandler [115] documented clay mineralogy and chemistry of Hamra soils in other sites along the coastal plain and found out somewhat smaller amounts of aluminum oxides—3–10% in comparison to 18% at our site, whereas the iron oxides are within the same range—0.8–5.4%.

3. Somewhat drier environmental conditions led to the development of calcic soil overtopping the Hamra, forming a stage III+–IV- calcrete (Fig 15) in areas that were topographically higher, with aerobic conditions and good drainage [116, 117]. Older, stable sand and aeolianite surfaces, truncated and covered by pedogenic calcrete, were also documented by Harel et al. [111].

4. Massive fluvial truncation of the sand and Hamra ridges by two alluvial channels that deposited coarse limestone gravels originating from the Judea Mountains (Unit 2; Fig 15). The channels display a local north-south flow direction and probably represent tributaries of the precursor of the adjacent Nahal Qana (Fig 1C), such as Wadi Asla, that flowed between the bottom of the western flanks of the Samaria Mountains and the Pleistocene/Holocene Red sand and loam exposure of Unit 1 at Jaljulia. The large gravels—up to 40 cm

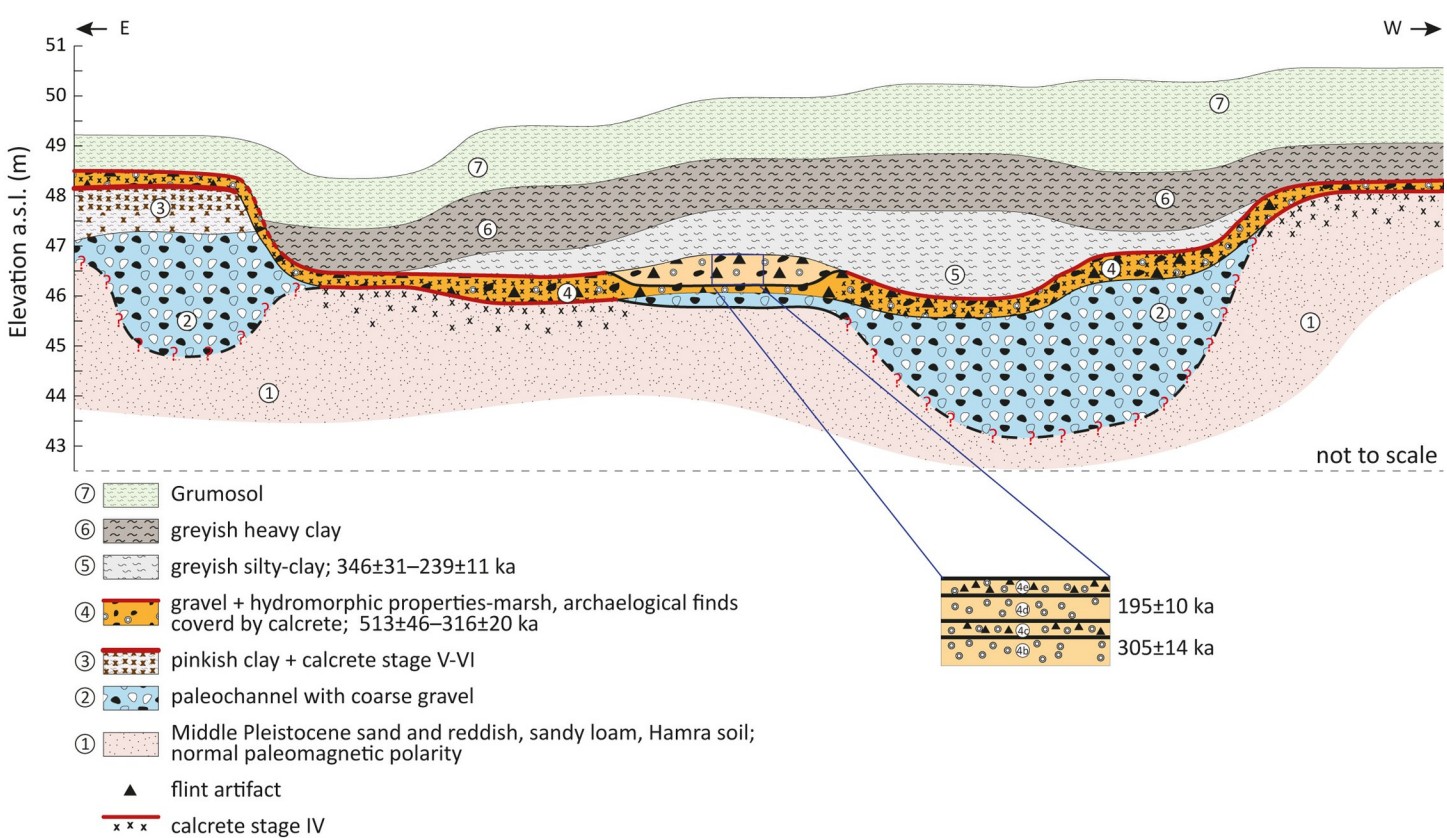

**Fig 15. Composite schematic reconstruction of the depositional history at the site of Jaljulia.**

in size, indicate energetic flows with estimated velocity of about 3 m/sec and critical shear stress of about 2300 dynes/cm$^2$ [118]. Since the full size of the channel is unknown the discharge cannot be calculated. The reddish matrix of Unit 2 is probably derived from the Terra Rosa soils mantling the slopes of the basin. These well-drained soils are composed of aeolian dust, reddish-brown iron and aluminium oxides, and clay minerals–mainly smectite and kaolinite [115]. His aluminum oxide amounts for mountainous Terra Rosa soils in central Israel, are slightly smaller—13% versus 19% at our site, whereas the iron oxides are slightly lower at our site– 4.4% versus 7% in average [115].

5. The exposure of Unit 2 gravels was followed by the deposition of a thick (>1m) reddish silty-clay (aeolian?) Unit 3. A long exposure time of Unit 3 to somewhat drier conditions with limited drainage and aerobic conditions resulted in the development of a Stage V–VI calcic soil [119, 120], prior to about 513±46 ky, displaying high Ca/Si ratio (4.3 on average) and low Si/Al ratio (1.7 on average; Table 2). The identification of this phase only in the topographically highest part of the site (Area A; Figs 2 and 15) implies that this stage was either confined to the most elevated localities, or, more logically, that most evidence for this phase at the lower topography, were eroded.

6. Fluvial erosion of Units 1–3 by energetic flows along the north-south channel and deposition of coarse chert gravels, up to 25 cm in size, of the Judea group (Unit 4) over the entire site (Fig 15) with a quartz-rich, clay matrix between 513±46 and 316±20 ky. This may indicate a fluvial pattern change from a more confined channel to a wider, braided channel which may have extended further eastward. Estimated velocity for the transport of these

gravels is about 3 m/sec and critical shear stress of about 1300 dynes/cm$^2$ [118]. The discharge for this unit cannot be calculated since the full size of the channel is unknown. The absence of carbonate gravels may be related to intensive chemical weathering in the following wetland phase.

The finer texture at the upper part of the unit indicates a dramatic decrease in flow energy, whereas the large amounts of aluminum and iron oxides, grey-yellowish colored gley, mottling and other reduction phenomena indicate post-depositional hydromorphic conditions related to a standing water body or wetland, either perennial or seasonal [121]. Large amounts of sand and dust-related quartz with high Si/Al ratio (3.6 on average), and negligible amounts of calcite (Table 2), also indicate water-rich environment, probably high groundwater levels. The presence of numerous artifacts on the gravels of this unit, correlates the human activity at the site with that water body. At the northwest, the scarce gravel in the relative elevated Area E and the minor hydromorphic evidence probably represent the margins of the water body and suggest that this part of the site was dry most of the time. The good preservation of artifacts at the northeastern part of the site (Areas B, G) (Figs 2 and 15) may suggest a lower degree of hydro-movement, whereas at the southern part of the site, Areas A (east) and C, water flow was probably stronger.

7. Development of a stage IV calcic soil (calcrete) over the entire site (Fig 15) and penetration of carbonates into the hydromorphic soil, represented by a second layer of carbonate coating on the larger gravels, indicating aerobic conditions and limited drainage, which can be related to a period of drying-up of the water body. The fact that well-developed calcic soil sealed the archaeological horizons suggests that the drying of the water body marked the end of the human activity at the site ca. 316±30 ky. This should be related to a climatic change into drier conditions and a significant contribution of carbonates into the region. In Area G, the absence of this calcic throughout the exposed sequence of archaeological horizons (Figs 2 and 15) may indicate continuous, shallow low-energy fluvial activity in a limited area at the center of the site while the other parts were already dry.

    The sedimentological analyses of Sub-units 4b and 4d in Area G show similar chemical composition with minor increase in aluminum, iron, and manganese oxides (Figs 2 and 15 and Table 2). Together with the fine, gravel-free texture, we suggest that the source of the sediments is local, probably derived from the exposed dry parts of the older wetland Unit 4. The somewhat younger luminescence age at the bottom of this sequence (Unit 4b, 305±14 ky) may support this idea. Alternatively, it may reflect stripping of these soils by younger streams, indicating that the human activity in Area G post-dates the activity at the rest of the site. Ongoing studies of the finds and the sediments from Area G are expected to shed more light regarding the role of Area G in relation to the other excavated localities at the site of Jaljulia. Notably, initial observations during field work indicated that the lithic technologies represented in Area G are not distinctly different from the other areas of the site, leading to their attribution to the Late Acheulian. If such observations are confirmed by final lithic analysis, the young chronology provided by OSL will have to be re-considered carefully (see below).

8. Truncation of the carbonate soil overlying unit 4 by flows that did not leave any evidence and deposition of fluvio-aeolian clay (Unit 5; Figs 2 and 15), dated by OSL to between 346 ±31 and 239±11 ky in the topographically lower areas (B, G, D and C). The massive structure suggests slow deposition associated with a temporary, shallow water body probably with a high concentration of sediments. Sedimentological analyses show quartz-rich sediments (73% on average; Table 2) with considerable amounts of aluminum and iron oxides suggesting wetter environment (Si/Al– 4.8 on average), with relatively poor drainage and

reduction processes. Charred old roots found in this unit indicate the presence of vegetation. This stage may represent temporary/seasonal, smaller, and shallow waterbody at the lower part of the site between the elevated areas A in the east and E in the west.

9.  Erosion and shifting of the drainage system to the eastern (Area A south; Fig 15) part of the channel and continuous aeolian-fluvial deposition of massive, grayish brown clays of Unit 6, probably covering the entire site. The prismatic structure of this unit, as well as the presence of slickensides large carbonate nodules in joints and some manganese concretions, suggest alternating dry and water-logged environments, which possibly reflect seasonal changes. Charred roots indicate vegetation cover.

10. Minor erosion at the eastern part of the site (Area A south) and continuous deposition of grayish-brown silty clay (Unit 7), which completely covered the entire site and is suggested to be mainly aeolian. The unit developed into a silty-clay vertisol (Grumosol).

## Environmental changes

Aeolian sand deposition and overtopping Hamra soil represents the initial cover of the coastal plain during mid-Pleistocene [113, 114], developing during a Mediterranean type of climate. The cover by a later calcic soil represents transformation to slightly drier hydrological conditions (Fig 15).

Dramatic fluvial period is represented by Unit 2 with large gravel, that cut into the sand and formed north-south trending channels, possibly within an existing north-south depression. This can be related to large events that can happen during transition from drier to wetter climate (Fig 15). In contrast to the Revadim Site, where channel dimensions could be documented and discharges were estimated [28], in the present study the full size of the channels (Unit 2 and 4) are unknown and therefore discharge cannot be estimated. Nevertheless, using the maximum gravel size we estimated the velocity and critical shear stress for the transport of these gravel sizes based on the relationships described in Costa [118].

While the original properties and deposition of Unit 3 are not clear due to the following intensive calcic pedogenesis as well as to its erosion by Unit 4, its composition (silty-clay) may indicate another aeolian ingression. The unit is completely covered by a well-developed calcic soil indicating another phase of drier conditions and relatively long exposure (Fig 15).

Unit 4 is a second fluvial period that transported large chert gravels and cut into the underlying Units 1–3. This high energy flood period may correspond to the colder and drier glacial period of MIS12. This fluvial phase was followed by development of temporary/perennial inland wetlands [121] dated to about 513–316 ky, between MIS12 and MIS9.2 (almost 200 ky), which include two glacial periods and two interglacial periods [122]. During the interglacial periods—MIS11 and MIS9.3 sea levels rose a few meters higher than present sea level, which caused ingressions of the coastline inland. Such ingressions during about 410 ky and 330 ky, were located at about 5.5 and 4km east of the present coastline, respectively, as inferred to by Harel et al. [111]. These rising and high sea levels caused incursion and deposition of marine sand inland and upstream blocking the outlets of the coastal rivers [121, 123]. Such phenomenon was documented at the narrow corridor of the Qishon Stream outlet, where such sand blockage forced the water of the stream into a marsh upstream the blockage [123]. The narrow corridor (0.3-1km wide) of the perennial Yarkon Stream to the coast—the main stem of which Wadi Qana is a tributary (Fig 1C), could have been blocked at least twice by marine sand during these interglacial periods, generating marshes upstream such as the Jaljulia wetland/marsh. This may have also led to higher local groundwater levels. These wetlands are characterized by hydromorphic soils, gley, and reduction environment. The generation of such inland marshes

were correlated to warmer and wetter climates in other sites in the region, whereas opening of these blockages and drainage of the marshes were caused by larger floods associated with cold and more arid periods [121, 123]. Regional wetter hydrological conditions can be correlated to a more humid period between about 290–350 ky as indicated by speleothem deposition in caves at the northeastern Negev and southern Judea mountains [124] as well as to the timing of Mediterranean sapropel S10 [125]. This wetter phase terminated by another calcic soil related to drier conditions that sealed the site and probably marked the end of the human occupation. Unit 5 (dated to about 346–239 ky; MIS9–8), as well as Units 6 and 7 represent thick, 2–4 m section of aeolian-fluvial deposits composed of silty-clays. Unit 5 may represent temporary/perennial shallow inland wetland with similar sedimentological characteristics as Unit 4 (Table 2 and Fig 15), whereas Units 6 and 7 are increasingly continental. This suggest that the north-south paleo-channel of Units 2 and 4 after the marsh periods was filled with these aeolian-fluvial deposits, forcing the channel to shift westward, west of the Jaljulia Red sand and loam exposure (see Fig 1C) to its present location.

**Summary.**   Unit 1 and the overlying Hamra soil represent the typical Middle Pleistocene environment–parent material and climate. Deposition of Units 2 and 4 indicate high-energy fluvial activity transporting large gravels through a north-south paleo0channel, whereas Unit 3 is probably aeolian with a drier environment. Wetland environments characterize Unit 4 and (partially) Unit 5 indicating wetter conditions during interglacial periods MIS11 and MIS9.3, as well as higher sea levels and coastlines shifts eastward, and resulting marine sand blockage of the Yarkon stream corridor to the sea.

The presence of three calcic soils with calcretes overtopping Units 1,3 and 4 can serve as markers for transitions to drier hydrological conditions with calcite accumulation due to limited drainage and/or increasing amounts of dust-sourced calcite (about 70% in average). The calcrete sealing Unit 4 indicates drying of the wetland, probably marking the end of human activity at the site.

Units 5, 6 and 7 indicate continental conditions of aeolian deposition which filled the paleo-channel, causing it to shift westward. The overlying grumusol developed during seasonal wetting and drying cycles (Fig 15).

## The age of the Late Acheulian site of Jaljulia

Several methods were used in the effort to provide an age estimation for the human activity at the site of Jaljulia. Analysis of the paleomagnetic stratigraphy of the profile in Area B showed a normal polarity for all tested samples, indicating that the whole sequence exposed on site was deposited during the Brunhes Chron, and is therefore younger than ca. 780 ky. Notably, similar paleomagnetic results were obtained for the Late Acheulian site of Revadim [19]. Absolute chronology at Jaljulia was obtained using TT-OSL and pIR-IR290 on quartz and feldspar grains, respectively, as well as ESR on quartz grains. Samples for all methods were collected together from the same locations within the sections, in order to ensure valid comparisons. Out of 11 samples presented in this paper, two were dated using all three methods (JAL 3, JAL 6), five using TT-OSL and pIR-IR290 (JAL 1, 7, 10, 20–21), three using TT-OSL and ESR (JAL 4–5, 9) and one using only TT-OSL (JAL 8).

Overall, the results of all three methods show a good correlation (Table 11). Sample results fit the stratigraphic order for the most part, with the two samples collected from Unit 1 marking the only exceptions. The ages obtained from these samples (~438–298 ky; Table 5) are very young, and do not fit the sedimentary attributes of the unit, as well as the geomorphological processes reflected in the site. High De values further support the assumption that these results are underestimations and reflect minimum ages.

The six samples collected from Unit 4 show a distinct spatial pattern, which might indicate chronological distinction between human occupations in the different excavated localities. Areas D and C present the oldest ages, and somewhat close ranges (~513–417 and ~444–428 ky, respectively), Followed by Area B (~367–316 ky). Most remarkable, however, are the results of the samples from Area G, yielding extremely young ages: ~310–305 ky for Sub-unit 4d, and ~200–195 ky for Sub-unit 4b. As the samples were collected from the silty Sub-units between archaeological horizons, they provide maximum and minimum ages for horizon G3 (~310 ky and ~195 ky, respectively) as well as maximum ages of for archaeological horizons G1–G2 (ca. 200–195 ka; Fig 2 and Tables 5, 6 and 11).

However, it must be stressed that these luminescence ages are very unusual for the Levantine Late Acheulian and are not yet supported by other dating methods. While the good correlation with ESR ages in other parts of the site grants us some confidence, the geomorphology of the site indicates that the youngest ages presented in Sub-unit 4b do not fit the deposition of the overlying Unit 5. Thus, in our view, further inspections into the reliability of the ages of Area G is required. Additional TL measurements of burnt flints and ESR age estimation of quartz particles from the sediments tested for TT-OSL and pIR-IR290 are currently underway, alongside a comprehensive analysis of the lithic assemblages from this locality. These will provide much needed additional criteria for evaluating the age of human activity in Area G, and until such data is available, we treat the suggested ages with caution.

The OSL ages from Areas B, C and D were tested against the ESR results and found to correspond, and at this stage of the research we would like to focus on these ages as the most reliable chronological estimations available at the moment for the site of Jaljulia.

Relying on the results of the absolute chronology from these three areas, we believe that the variability in the ages of Unit 4 reflects a composite, multi-phased deposition, in which three distinct phases of human activity might be evident. Areas C and D represent the earliest of the three, placing the human activity within the MIS 11, roughly 513–417 ka. The second phase is represented by Area B (and possibly also by horizon G3 in Area G), indicating human activity at the site also during MIS 9. The possibly youngest phase is seen in horizons G1-G2, as luminescence ages place them well within MIS7 and possibly early in MIS6. However, as stated above, the younger ages will have to be reconsidered in the light of more ages and the comparative lithic analysis. Nevertheless, we offer this data as food for thought for the scientific community dealing with the study of the Levantine Lower Paleolithic.

All human activity on site is associated with a long period of wetlands/marsh environmental conditions (seasonal or perennial), which followed the deposition of Unit 4 gravels, sometime during MIS12. Clear evidence for a younger phase of geological activity related to Unit 4 is shown in Area G, where the detailed stratigraphy indicates a continuous fluvial activity, with

**Table 11. Summary of absolute dating results for Unit 4 (in bold) and Unit 5 according to locality.**

| Sample ID | Locality | Unit | TT-OSL | pIR-IR290 | ESR (Al) | ESR (Ti-Li) |
|---|---|---|---|---|---|---|
| JAL-10 | B | 5 | 239±11 | 231±13 | | |
| JAL-9 | **B** | **4 (high)** | **316±20** | | **330 ± 30** | **NC** |
| JAL-8 | **B** | **4 (low)** | **367±24** | | | |
| JAL-6 | C | 5 | 299±14 | 284±19 | 220 ± 40 | 335 ± 50 |
| JAL-5 | **C** | **4** | **428±25** | | **444 ± 30** | **446 ± 30** |
| JAL-3 | D | 5 | 346±31 | 335±28 | 378 ± 70 | 320 ± 30 |
| JAL-4 | **D** | **4** | **513±46** | | **462 ± 60** | **417 ± 50** |
| JAL-21 | **G** | **4d** | **195±10** | **200±11** | | |
| JAL-20 | **G** | **4b** | **305±14** | **310±16** | | |

alternating gravel (4e, 4c, 4a) and silty clays (4d, 4b) deposition. Sedimentary properties correlate the earliest Sub-phase 4a with the fluvial activity that deposited the gravels of Unit 4 throughout the site (Table 2), implying that Sub-units 4b–4e represent younger fluvial depositions, possibly marking a reduction in the size of the water body and a change in environmental and hydrological conditions.

## Jaljulia in a regional perspective

The site of Jaljulia is a unique point in the map of Late Acheulian sites in the southern Levant. Its magnitude and its position in the central coastal plain of Israel sheds light on the potential this region still holds for the research of the Levantine Acheulian. The geology and chronology of the site indicate that it shares the basic attributes of many of the Late Acheulian sites in the southern Levant, demonstrating repetitive occupations associated with a perennial or seasonal water source. As a fresh water source is likely to support lush vegetation as well as prey species, the advantages of such a locality for Late Acheulian hominins are clear. At the site of Jaljulia, another possible advantage was the cherty composition of Unit 4 gravels, which allowed for a rich, accessible raw material for flintknapping.

The lithic industries, presented here briefly through the assemblage from Area B, can be described as typical of the Levantine Late Acheulian, and bear great similarities to the assemblages from the Late Acheulian sites of Revadim, Kefar Menachem West, Berekhat Ram and Eyal 23 [9, 17, 26, 33, 35, 126]. The flake industries demonstrate the application of a combination of core technologies, including three main reduction sequences: 1. The production of large, medium, and small sized flakes from one-, two- or multi-striking platform cores; 2. The production of predetermined items from prepared cores; and 3. The production of very small flakes from core-on-flakes in a mode of lithic recycling.

The percentage of shaped items, including the bifaces category, is rather high and comparable mainly to the site of Revadim where this category constitutes 4.9–12.9% [35], whereas in other sites such as Berekhat Ram and Kfar Menahem West, lower frequencies of shaped items are noted (4–7% and 7.8%, respectively; [9, 17]). Such a high percentage could imply a wide range of activities conducted on-site or high intensity of production and use. As such differences might also be influenced by artifact classification procedures and standards of different research groups it is worth noting that the two sites of Jaljulia and Revadim were studied by the same research group. The abundance of bifaces, as well as typical maintenance and shaping debitage, is associated with the production of these items on-site, possibly from the cherty gravels of Unit 4. Preliminary use-wear analysis suggested the use of some of the bifaces for pounding and breaking bones and other related activities [99]. In addition, the use of some of the handaxes as cores after their discard as tools was suggested to represent a particular approach for the production of predetermined flakes [42]. All suggest long sequences of handaxe production, maintenance, use and recycling on-site.

As for understanding the broader chronological and behavioral implications of this study regarding the probable overlap in time between the Late Acheulian and the AYCC, or more particularly between the sites of Jaljulia and Qesem Cave which are only six km apart, we feel more data is needed. The most reliable ages from Areas C and D of Jaljulia are clearly earlier than the established AYCC chronology of Qesem Cave, and thus it seems that they represent Late Acheulian occupations which predate the AYCC at Qesem. Nevertheless, the ages suggested for Area B seem to be contemporaneous with the chronological range represented at Qesem Cave, and those of Area G even seems to be slightly younger than the range of Qesem Cave chronology. While the ages of Area G still need to be further investigated, we consider the chronology of Area B reliable due to the good correlation between TT-OSL, feldspar and

ESR results. As the lithic assemblage from Area B is clearly Late Acheulian in nature, the overlap with the chronological range suggested for the AYCC of Qesem Cave indeed might imply some contemporaneity between the occupations of the two cultural groups in a close geographic region.

These observations are intriguing and hold significant potential for enhancing our understanding of cultural and behavioral transformations in the Levant during these significant times in human cultural and biological evolution. We intend to further pursue and study these aspects as the processing and analyses of the finds advance.

The possibility that at least three phases of occupation are represented on-site, supported by absolute ages, provide a unique opportunity to study diachronic changes in material culture within the Late Acheulian. Even though each of the excavated localities could in fact represent a palimpsest of short-term occupations, the study of each locality can potentially teach us about subtle changes in lithic traditions and preferences through time. Furthermore, the study of the assemblage from Area E is expected to provide a glimpse into human activity in dry environments slightly distant from, and at the margins of the waterbody dominating the rest of the site.

## Concluding remarks

A recently discovered, large-scale Late Acheulian site was excavated at the town of Jaljulia, in the central coastal plain of Israel. Multidisciplinary study of the site indicates repeated human activity in a water rich environment, in immediate proximity to potential secondary source of stone for flintknapping. Geomorphological analyses indicate a change in depositional environment, with wetter climate conditions and higher sea levels during the deposition of Units 4 and 5 (partially), correlating to MIS11 and MIS9.3, which caused coastline shifts to the east, resulting in a blockage of the Yarkon stream corridor to the sea.

Analysis of the flint assemblage from Area B shows the classic attributes of Late Acheulian industries, including flake production from cores and cores-on-flake, and production of predetermined flakes from prepared cores and from discarded bifaces. Absolute ages provided a chronological frame of ca. 500–300 ky for the main human activity on-site, followed by a younger phase of activity, the chronology of which is still under investigation. Implied chronological overlap between the Late Acheulian of Jaljulia and the Acheulo-Yabrudian Cultural Complex of Qesem Cave which are only 6 km apart, opens a venue for the study of possible connections and technological continuation between the two hominin lineages occupying the region during the Middle Pleistocene. Such lines of investigations are at the center of our ongoing studies, and the site of Jaljulia is expected to shed more light on the nature of human activity, adaptation, and creativity in the southern Levant during Late Acheulian times.

## Supporting information

**S1 Fig. Jaljulia, Area A.** From the center: a) Air view of Area A at the end of the excavation. b) The eastern section showing the sedimentary units detailed in chapter 3.1. The scale is 1 m. c) The eastern excavation squares. The scale is 50 cm. d) The main excavation of Area A. The scale is 50 cm. e) a closeup of Unit 4, containing lithic artifacts, in the main excavation of Area A. Each section of the scale marks 10 cm.
(TIF)

**S2 Fig. Jaljulia, Area B.** From the center and counterclockwise: a) Air view of Area B at the end of the excavation. b) Geological section showing the sedimentary units described in chapter 3.1. c) a closeup of the section. d) Area B at the end of the excavation. The scale marks 50

cm. e, f) closeups of Unit 4 displaying high frequencies of lithic artifacts. Scales are 20 cm (e) and 10 cm (f).
(TIF)

**S3 Fig. Jaljulia, Area C.** From the center: a) Air view of Area C at the end of the excavation. b) Area C, after removing the archaeological horizon and digging into Unit 4. c) A close up of the section of Area C, displaying the sedimentary units described in chapter 3.1. d) The top of Unit 4 in Area C. e) A close up look of the archaeological horizon at the top of Unit 4.
(TIF)

**S4 Fig. Jaljulia, Area D.** From the center and counterclockwise: a) Air view of Area D at the end of the excavation. b) Area D and the southern section. c) Area D and the southern section, presenting the sedimentary units described in chapter 3.1. d) A close up look at the top of Unit 4, containing high frequencies of lithic artifacts. e) The archaeological horizon at the top of Unit 4, as was exposed in Area D. f-h) close up views of the archaeological horizon. (f) is showing one of the few faunal remains preserved on site. Scales are 50 cm (g) and 5 cm (h).
(TIF)

**S5 Fig. Jaljulia, Area E.** From the center: a) Air view of Area E at the end of excavation. b) The southern section, displaying the sedimentary units described in chapter 3.1. c) The archaeological horizon as was exposed in Area E. The scale (at the center) marks 50 cm. d, e) A close up look of two of the find-clusters exposed in Area E. Scale is 5 cm.
(TIF)

**S6 Fig. Jaljulia, Area G.** From the top left: a) Air view of Area G at the end of the excavation. b) The southern section, displaying Unit's 4 sub-division as described in chapter 3.1. c) Area G during the exposure of sub-unit 4e. The scale marks 50 cm. d) A close up look on archaeological horizon G1. The scale is 5 cm. e) A close up look at archaeological horizon G2. The scale is 5 cm. f) Sub-unit 4c (= archaeological horizon G3). The scale marks 20 cm. g) A close up look at archaeological horizon G3. The scale is 5 cm. h) Sub-unit 4a (= archaeological horizon G4). i) A close up look at archaeological horizon G4. The scale marks 20 cm. j) Area G after the removal of sub-unit 4a and the exposure of the underlying conglomerate of Unit 4.
(TIF)

**S7 Fig.** Orthogonal comparative display of representative results of AF (a) and Thermal (b) demagnetizations conducted on samples from horizon GAL 2.
(TIF)

**S1 Table. U, Th, K contents in the sediments of Jaljulia.**
(DOCX)

**S2 Table. Detailed results of the sedimentary analyses of the Jaljulia samples showing main elements frequencies.** *IC measurement.
(DOCX)

**S3 Table. Detailed sedimentary analyses results for the samples from Jaljulia, displaying main minerals frequencies.** *IC measurement.
(DOCX)

## Acknowledgments

The Authors wish to extend their gratitude to the excavators and field supervisors: N. Solodenko, B. Efrati, Y. Keidar, A. Agam, E. Assaf. We also thank the students of the Department

of Archaeology in Tel Aviv University for their assistance during the excavation. LKH wishes to thank F. Natalio and Z. Stepka (Weizmann Institute) for performing the SEM-EDS analyses.

## Author Contributions

**Conceptualization:** Maayan Shemer, Ran Barkai.

**Data curation:** Maayan Shemer, Noam Greenbaum, Nimer Taha, Lena Brailovsky-Rokser, Yael Ebert, Ron Shaar, Christophe Falgueres, Pierre Voinchet, Naomi Porat, Galina Faershtein, Liora Kolska Horwitz, Tamar Rosenberg-Yefet, Ran Barkai.

**Formal analysis:** Noam Greenbaum, Nimer Taha, Yael Ebert, Ron Shaar, Christophe Falgueres, Pierre Voinchet, Naomi Porat, Liora Kolska Horwitz, Tamar Rosenberg-Yefet.

**Investigation:** Maayan Shemer, Noam Greenbaum, Christophe Falgueres, Naomi Porat, Ran Barkai.

**Methodology:** Maayan Shemer, Naomi Porat, Galina Faershtein, Ran Barkai.

**Project administration:** Maayan Shemer.

**Resources:** Maayan Shemer, Ran Barkai.

**Software:** Lena Brailovsky-Rokser.

**Supervision:** Ran Barkai.

**Writing – original draft:** Maayan Shemer, Noam Greenbaum, Christophe Falgueres, Naomi Porat, Tamar Rosenberg-Yefet, Ran Barkai.

**Writing – review & editing:** Liora Kolska Horwitz.

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
