## [Decision Letter · Decision Letter 0]

28 Feb 2022

PONE-D-21-39134Late Acheulian Jaljulia – Early Human Occupations in the Paleo-landscape of the Central Coastal Plain of IsraelPLOS ONE

Dear Dr. Shemer,

Thank you for submitting your manuscript to PLOS ONE. After careful consideration, we feel that it has merit but does not fully meet PLOS ONE’s publication criteria as it currently stands. Therefore, we invite you to submit a revised version of the manuscript that addresses the points raised during the review process.

All the reviewers agree that this site will be a worthy addition to our knowledge about the Levantine Acheulian. However, they also point out some important structural and substantive issues that should be dealt with before it is published.

Reviewer 1 and 2 think consider the claims about the lithics and comparisons with Qesem unwarranted given the state of the analysis. Please consider revising to focus more on the chronology and dating, and less on the behavioral interpretation derived from the lithics. Reviewers 2 and 3 also point out some issues in relating the deposition of the sediments and of the lithics, which should be addressed in the revision.  

We look forward to receiving your revised manuscript.

Kind regards,

Radu Iovita

Academic Editor

PLOS ONE

Journal Requirements:

2. In your Methods section, please also include the text regarding your field permit for this study that you have entered in the Ethics Statement on the submission form.

Additionally, in your manuscript, please provide additional information regarding the specimens used in your study. Ensure that you have reported specimen numbers and complete repository information, including museum name and geographic location.

For more information on PLOS ONE's requirements for paleontology and archaeology research, see " ext-link-type="uri" xlink:type="simple">https://journals.plos.org/plosone/s/submission-guidelines#loc-paleontology-and-archaeology-research."

Reviewers' comments:

Reviewer's Responses to Questions

**Comments to the Author**

1. Is the manuscript technically sound, and do the data support the conclusions?

Reviewer #1: Yes

Reviewer #2: Partly

Reviewer #3: No

2. Has the statistical analysis been performed appropriately and rigorously? 

Reviewer #1: No

Reviewer #2: N/A

Reviewer #3: N/A

3. Have the authors made all data underlying the findings in their manuscript fully available?

Reviewer #1: Yes

Reviewer #2: Yes

Reviewer #3: Yes

4. Is the manuscript presented in an intelligible fashion and written in standard English?

Reviewer #1: Yes

Reviewer #2: Yes

Reviewer #3: No

5. Review Comments to the Author

Reviewer #1: This contribution summarizes the industrial and chronological data from The site of Jaljulia is a unique point in the map of Acheulian sites in the southern Levant. The information provided deserves publication due to the relevance of the site and the initial registration data from it, mainly in relation to the chronology, assembal¡ge, and site evolution.

In some aspects, the paper is worth enough to be published, but at the same time, some others need to be reviewed or discussed. Please provide a reflection to the following questions:

1) I consider that the main cope of the article is somehow missing. Is it a presentation of the chronology? A general description of the industry? the general context of the site? I encourage the authors to define clearly from the beginning the aim of it.

2) Along with the data presentation, the authors constantly include the mention of its assignation to the Late Acheulian without a clear definition of the main features of this period. It is crucial to have a clear understanding of this technocomplex/industrial expression... Perhaps it would be better to define in a sentence what elements can classify an assemblage as Late Acheulian or if the author's criteria are based on chronology, raw material selection, or other aspects.

3) When some of these possible features were analyzed¡, the author indicates that in relation to the flaking productions, cores present as an innovative element the control of volume. But bifacial productions also have this volumetric conception. Please provide us with more information about flaking methods perhaps by using their diachritic examples.

4) What is the meaning of such an amount of debris (56,3%) in the site's context? Please check figure captions and the types used on the tables.

5) In relation to the classification of cores, the authors merely describe them by a simply morphological approach. Usually, the reduction process could define the existence of a real concept in the flaking methods used, and the final expression of core morphologies hide it. Please try to describe them and not just to define "morphologies" are they discoid? multiplatform? levallois?

6) In relation to this, What is the difference between the levallois and protolevallois documented cores? Volumetrically some pieces of figure 9 are levallois.. the existence of discoid and other volume core results must be taken carefully...

7) When the authors used discoid core categories, were they real "Boeda's" discoid cores? Terradas discoid definition does not analyze final discoid products that are the tricky and key aspects to be analyzed from cores: knifes pseudolevallois flakes and points, quadrangular flakes, etc. By the way, the discoid cores presented in figure 9 seem to be exhausted cores.

8) This could also happen with the author's "handaxe with preferential flake scars" that are not clearly defined by using a technological reading (order in negatives).

9) For the conclusions, it would be appropriate to establish a general context of the Acheuliand and particularly to the last episodes, in the European context and perhaps also in Asia, to evaluate the relevance of the data provided by this site

In ocnclusion, the paper is relevant and, after some review, could be published. The regional importance of the regional sites is crucial for understanding technocultural evolution on small and large scales.

Reviewer #2: The core of this paper is a presentation of the geology and dating of the Jaljulia site in Israel. There is also some information on the lithics and the minimal bone assemblage presented. I think the paper should be published for the geology and dating. I don't suspect there is more that could be done with the fauna. For the lithics, they note that they are still under study, and what is presented here I would view as extremely preliminary. As such, I think maybe the focus of the paper in the introduction and in much of the discussion and conclusion is a bit too heavy on the lithics. The authors seem to want to put the site into comparison with, for instance, Qesem Cave but it really seems premature given what is presented here.

Further, it seems like one of the main issues will be reconciling the dating of the archaeological horizons. If all of the dates are accepted at face value, it seems that Unit 4 was exposed and occupied for a very long period of time (i.e. across multiple MIS cycles). This seems unlikely, no? So I had a hard time making sense of that.

And in terms of site formation processes, the geology seems quite good, but on the archaeology side it is a bit weaker. On the archaeology site there are some general statements about orientations or preservation but no concrete data. Clearly Unit 4 was deposited in a high energy situation. And the faunal analysis argues that the fauna are derived. The lithic analysis says that some of the material might be derived, but argues that most of it is in situ. But the support for the latter is really missing. I think they could be right. The layer could be deposited in high energy but the archaeology not. But this really needs to be shown. I feel like what is really needed is a better site formation study from the lithics and a better effort to resolve the differing dates from different parts of the site from a lithics perspective (i.e. do the lithics from each area all look the same or do they differ in some interesting ways).

I agree with the authors that this site will make a nice additional data point for the late Acheulian (however, the near complete lack of fauna and no evidence of spatial structure mean some significant limits to what can be said of human behavior). I would though reorient the paper a little less around the lithics and questions of AYCC, Late Acheulian, MP etc. given that in this paper they are clearly secondary to the dating and geology.

Some more detailed comments:

In this introduction, the notion that lithics equal people is pretty clear. For me, I don't think it is clear that lithics equal people, but I realize that this is not the prevailing view of those doing Levantine Paleolithic. Still, maybe some effort could be made to discuss the lithics as a component of human groups that occupied the Levant during these various periods. Whether changes are driven by demographic changes (incoming groups) or represent local changes is mostly (entirely?) unclear throughout this whole period.

Line 109 - So Late Acheulian is a chronological phase or a technological phase?

Line 126 - Coming back to my comment above about lithics equal people, the discussion in this paragraph is appropriate.

The manuscript could use a better section numbering system so that the hierarchy is better understood. For instance, there is "dose rates" on line 279 and 316. I know they related to different dating methods, but it could be made more clear by linking them with section numbers.

Line 331 - 7 is the oldest or youngest? [I see now it must be the youngest - but state it here]

Table 3 - Is the size of the lithic assemblage all lithics greater than 5mm (the screen size in most cases)? Maybe state this somewhere. Area G is NA for what reason? And G1-G5 are layers? Is this explained somewhere? [I see now on Line 465 - maybe reference to this needs to go earlier]

Line 455 - best state of preservation is based on what?

Line 457 - No data on orientations?

Line 458 - No data on ratios of chips to larger artifacts?

Line 459 - How do limestone slabs mean that there is little post-depositional movement?

So Unit 1 is older than Unit 7, but horizons G1 and G2 are younger than horizons G3 and G4. And 4a is older than 4e?

Line 470 - No data for this?

Any explanation for why faunal preservation is only in Area D?

Actual data on artifact preservation by area would be nice (and ratio of small to large finds).

I am not a dating expert, but clearly there are a wide ranges of dates here each with fairly wide error ranges. I think a figure summarizing the results would be good.

Line 591 - Why does Area B have a date? Is the dating by area or for the artifact find horizon?

Table 7 - I don't get what modified and non-modified base flakes are. The base is the platform? And how is a non-modified base flake different from a primary element flake? And if the flake has a lip, it goes in its own category exclusive of these other categories? Shaped items is bifaces and also scrapers etc.? I don't know but there is no lithics method in the methods section. Shouldn't there be something about the methods and terms somewhere here? It sort of seems like the definitions come after the results.

Line 659 - What are retouched flakes that are not scrapers, denticulates or notches?

I am wondering what the significance is of the bone size plot? I don't understand it either (a ternary diagram with percentages for length, width and thickness). Not sure how to interpret this.

Line 699 - I guess this is the point of the size diagram, but I still don't quite get it.

As I reach the end of the paper, I find the summary of the dates. Maybe a figure is not needed as I gather all of the dates are reported on the schematic section? Maybe not?

Does it really make sense that Unit 4 was exposed over the course of several MIS stages and yet its occupation in each case always represents wetlands/marsh. Seems likely that it represents less time than that.

Line 929 - Says that more data are needed. But then the authors report a number of highly speculative things with no supporting data (e.g. scrapers with scalar retouch, blade production).

Line 963 - I didn't get that there were three phases of occupation supported by geomorphology. The dates maybe, but I didn't see that in the geomorphology.

I would delete the paragraph on Line 971.

Line 985 - Again, I don't see support for these speculations about links to Qesem Cave.

Reviewer #3: This article presents an important new site and is thus is very appropriate for publication in PLOS One. But the current version is fairly incoherent, apparently shaped more by the authors' preconceptions rather than the actual data they are reporting. Adding to this problem, the geological analysis, which is absolutely critical to this presentation is very difficult to follow and does not seem to be grounded in a serious effort to engage with fluvial geomorphology. The lithic analysis is adequate for a preliminary presentation and the faunal assemblage is very limited but well presented. For their revisions the authors need to do the following:

1. Remove all discussion of why the correct age is in any way known in advance of dating. There is simply no archaeological or geomorphological support for such a position and it stands in contrast to excellent analytical work. The claim that the OSL ages are problematic due to high De measures requires full substantiation and seems to directly contradict the earlier statement that there is high confidence in the ages-- which moreover include TT-OSL on quartz and IRSL on feldspars, both methods that work for sites much earlier than the one reported on here. I would point to the large number of dating samples as further strengthening the overall confidence of where in time this site is situated. I found the agreement between the ESR and OSL for Area C particularly impressive. The authors need to at least consider the possibility that this site (Unit 4) formed over a long period of time, as appears to be indicated by the age determinations.

2. The integration of the geology into the coastal geomorphology is very weak. I do not really comprehend why so much space is given to chemical analysis where we know that all fluvial derived sediments have the same source in the neighboring hills. There needs to be much more serious engagement with fluvial processes, as has been done in paleoclimatic reconstructions at Revadim (Malinsky Buller et al. 2011). One particularly interesting issue is how to interpret periods of incision, indicated here by what appear to be two channels in a unit underlying the site. This could be evidence of high rainfall but it is equally important to consider the effects of sea level on fluvial dynamics. What I most wanted to understand was what the authors think was the depositional context for the archaeological occupation. In the generalized cross section it appears that Unit 4 fills a basin that actually cuts into the earlier deposits, are we looking a broad fluvial channel? Finally, from the schematic sections there is reason to question whether Unit 4 actually represents a single depositional unit. Could we be looking at a braided stream that evolved over a long period? This would be logically consistent with the dating results. From the images it appears that the artifacts accumulated during the deposition of Unit 4, this should be made more clear-- it might be useful to have some vertical data on artifact distribution by spit in the different areas. Essentially it looks like what might be taking place is a shift from two incising channels (without associated archaeological material) to a braided stream that flowed through the area and is associated with dense archaeological accumulation. For assessing the association between the dated sediments and the artifacts is critical as the age determinations might actually record the time of sediment deposition rather than time of initial artifact deposition.

3. Adding to these problems is the visual presentation of data. Although Figure 1 is useful we also need a drawing of the excavation areas that is easy to key into Figures 2 and 15. I also did not understand some of the elements of Figure 15, for example how do the authors know that Unit 1 underlies the entire site if it was only reached in the central area? I am assuming that the energy of Unit 4 was responsible for the erosion into the underlying strata but this is not clear. I would have been interested to see a map positioning the site relevant to known paleo-drainage systems. This article appears to dismiss the system of kurkar ridges often included in scenarios for the development of the coastal plain, it would be interesting to also include on a map the location of known kurkar ridges.

I found there was consistent use of imprecise language not appropriate for a scientific publication. For example, line 176 "using standard methods for prehistoric archaeology in the Levant" , standard for who? This implies that there were no reasonable alternatives where this is not the case. The authors need to justify their method, not appeal to some alleged consensus methodology. Line 125 "this approach is mostly viewed with caution", by who? why? The authors need to ground assertions.

In sum, this article presents an important new site and an excellent suite of age determinations. There is no question that this is a fundamental contribution to the prehistory of the Levant and that this article will serve as the foundation for follow-up publications, particularly on the massive lithic assemblage. This is a clear illustration of the successful application of excavation in advance of construction, for which the authors are to be congratulated. However, the presentation needs to be integrated into a geomorphological scenario and all statements dismissing the age determinations based on archaeological assumptions must be removed.

6. PLOS authors have the option to publish the peer review history of their article (what does this mean?). If published, this will include your full peer review and any attached files.

Reviewer #1: No

Reviewer #2: No

Reviewer #3: No

---

## [Author Response · Author response to Decision Letter 0]

11 Apr 2022

Dear editor and reviewers,

Following your advice, we changed the focus of our manuscript so a bigger emphasis is put on our geological reconstruction and absolute chronology. Accordingly, the text was heavily revised, as some sections, containing some interpretations based on the lithic assemblage, were removed while others, correlating our local geology with the global climate and sea levels, were added. In addition, some of our figures were modified (Figs 1, 13, 15) to better present our data. 

We thank you for your time and for your helpful comments.

---

## [Editor Report · Decision Letter 1]

13 Apr 2022

Late Acheulian Jaljulia – Early Human Occupations in the Paleo-landscape of the Central Coastal Plain of Israel

PONE-D-21-39134R1

Dear Dr. Shemer,

We’re pleased to inform you that your manuscript has been judged scientifically suitable for publication and will be formally accepted for publication once it meets all outstanding technical requirements.

Kind regards,

Radu Iovita

Academic Editor

PLOS ONE

Additional Editor Comments (optional):

Thank you for following the reviewers' suggestions and focusing the paper on the geological and chronological aspects.  
---

## [Editor Report · Acceptance letter]

19 Apr 2022

PONE-D-21-39134R1 

Late Acheulian Jaljulia – Early Human Occupations in the Paleo-landscape of the Central Coastal Plain of Israel 

Dear Dr. Shemer:

I'm pleased to inform you that your manuscript has been deemed suitable for publication in PLOS ONE. Congratulations! Your manuscript is now with our production department. 

Kind regards, 

on behalf of

Dr. Radu Iovita 

Academic Editor

PLOS ONE